# Genome-Wide Analysis, Modeling, and Identification of Amino Acid Binding Motifs Suggest the Involvement of *GH3* Genes during Somatic Embryogenesis of *Coffea canephora*

**DOI:** 10.3390/plants10102034

**Published:** 2021-09-28

**Authors:** Hugo A. Méndez-Hernández, Ana O. Quintana-Escobar, Miguel A. Uc-Chuc, Víctor M. Loyola-Vargas

**Affiliations:** Plant Biochemistry and Molecular Biology Unit, Yucatan Scientific Research Center, Calle 43, No. 130 x 32 y 34, 97205 Mérida, YUC, Mexico; hugo.mendez@cicy.mx (H.A.M.-H.); ana.quintana@estudiantes.cicy.mx (A.O.Q.-E.); miguel.uc@cicy.mx (M.A.U.-C.)

**Keywords:** auxins, docking, *Coffea canephora*, *GH3*, plant growth regulators, somatic embryogenesis

## Abstract

Auxin plays a central role in growth and plant development. To maintain auxin homeostasis, biological processes such as biosynthesis, transport, degradation, and reversible conjugation are essential. The *Gretchen Hagen 3* (*GH3*) family genes codify for the enzymes that esterify indole-3-acetic acid (IAA) to various amino acids, which is a key process in the induction of somatic embryogenesis (SE). The *GH3* family is one of the principal families of early response to auxin genes, exhibiting IAA-amido synthetase activity to maintain optimal levels of free auxin in the cell. In this study, we carried out a systematic identification of the *GH3* gene family in the genome of *Coffea canephora*, determining a total of 18 *CcGH3* genes. Analysis of the genetic structures and phylogenetic relationships of *CcGH3* genes with *GH3* genes from other plant species revealed that they could be clustered in two major categories with groups 1 and 2 of the *GH3* family of *Arabidopsis*. We analyzed the transcriptome expression profiles of the 18 *CcGH3* genes using RNA-Seq analysis-based data and qRT-PCR during the different points of somatic embryogenesis induction. Furthermore, the endogenous quantification of free and conjugated indole-3-acetic acid (IAA) suggests that the various members of the *CcGH3* genes play a crucial role during the embryogenic process of *C. canephora*. Three-dimensional modeling of the selected CcGH3 proteins showed that they consist of two domains: an extensive N-terminal domain and a smaller C-terminal domain. All proteins analyzed in the present study shared a unique conserved structural topology. Additionally, we identified conserved regions that could function to bind nucleotides and specific amino acids for the conjugation of IAA during SE in *C. canephora*. These results provide a better understanding of the *C. canephora GH3* gene family for further exploration and possible genetic manipulation.

## 1. Introduction

Somatic embryogenesis (SE) has been under study for a little over 60 years [1]. During this period, empirical studies have documented a large part of the molecular, biochemical, cellular, and physiological mechanisms involved in triggering the signal that converts a somatic cell into an embryo [2]. However, an essential part of the signaling mechanism in the induction process of SE is still unknown. Auxins play a central role in this signaling [3].

Auxin, in its free form, acts as a signal molecule [4,5]. Auxins play an essential role during cellular and vascular differentiation, phototropism, plant morphology, and somatic and zygotic embryogenesis [3,6,7]. Their endogenous levels are controlled by various regulatory mechanisms such as de novo biosynthesis, transport, irreversible degradation, and reversible inactivation mediated by indole-3-acetic acid (IAA) conjugation [8,9]. Auxin homeostasis involves the leading families of genes that are expressed in response to auxins [10], including *AUXIN/INDOLE-3-ACETIC ACID* (*AUX/IAA*), *SMALL AUXIN-UP REGULATED RNA* (*SAUR*), *GRETCHEN HAGEN3* (*GH3*), and *AUXIN RESPONSE FACTOR* (*ARF*) [11]. These families of genes have been designated as early auxin-responsive genes.

The first report of the *GH3* genes was in *Glycine max* as an auxin-induced gene in etiolated hypocotyls [12]. Its mRNA is transcriptionally induced within 5 min of auxin treatment [13]. By sequence analysis and three-dimensional structure prediction of proteins, *GH3s* were found to belong to the acyl adenylate-forming firefly luciferase superfamily. The GH3 proteins catalyze a two-step reaction, where an acyl substrate (growth regulator) first gets adenylated. Then, a nucleophilic attack on the amine group of the amino acid results in the formation of a conjugated molecule of the auxin and the amino acid [14].

Most GH3 proteins regulate optimal levels of auxin in the cell, maintaining their homeostasis [15]. These enzymes catalyze reactions using indole-3-acetic acid (IAA) [16]. However, it has been reported that some GH3 proteins show activity with other plant growth regulators, such as jasmonic acid (JA) and salicylic acid (SA) [17]. Nonetheless, the activity of many GH3 proteins has not been characterized yet, and the nature of their substrates remains unknown. Recently, there is evidence that IAA is also able to form high molecular mass amide conjugates with proteins in pea [18]. The number of GH3 members varies greatly; e.g., there are 19 in *Arabidopsis thaliana* [19], 13 in *Oryza sativa* and *Zea mays* [20], eight in *Vitis vinifera* [21], 15 in *Solanum lycopersicum* [22], and 66 in *Brassica napus* [23], among others.

Currently, the GH3 proteins of *A. thaliana* have been classified into three major clades (I, II, III) according to their phylogenetic relationship and substrate specificity tests [16]. In group I, *AtGH3.11*/*AtJAR1* (*JASMONATE RESISTANT 1*) catalyzes the ATP-dependent conjugation between isoleucine and JA to yield jasmonyl-isoleucine (JA-Ile), the active form of jasmonate hormone [15]. Group II enzymes catalyze the formation of conjugates between auxins (mainly IAA) and amino acids, working as a regulatory mechanism to maintain auxin homeostasis by controlling the amount of free active auxin in the cell [16,24]. When the auxin is conjugated, the amino acid with which it binds determines if the molecule is degraded by an oxidative route or stored in the cell to be reused and converted into a signaling compound [25]. In group III, *GH3.12*/*PBS3* (*AVRPPHB SUSCEPTIBLE 3*) catalyzes the conjugation between 4-substituted benzoates and amino acids [26].

In the case of SE, the signals that lead to changes in the genetic program require the participation of several metabolic pathways. Auxins play a central role during the pretreatment and induction of the embryogenic process [3]. For example, in carrot cell cultures, it was observed that the use of 2,4-dichlorophenoxyacetic acid in the culture medium stimulates the accumulation of endogenous IAA and amide conjugates [27]. During the induction of SE in *Coffea canephora*, it was determined that the addition of an exogenous auxin during pretreatment of the source of the explants produces a sharp increase in the content of free IAA and some of its conjugates, such as IAA–Glu and IAA–Ala [28,29]. These results suggest that endogenous IAA accumulation appears to be essential for the changing of a cell’s fate before the beginning of the embryogenic pathway. The free IAA decreased within a few hours after the induction of the SE process and increased the transcripts of *GH3.1* and *GH3.17*. This increase correlates with the increase in the IAA–Ala and IAA–Glu conjugates [28]. It is also related to an essential expression of *GH3* family genes. This relationship suggests that an optimal balance between free IAA and its conjugates is necessary to allow expression of SE-related genes [28]. It has been determined that more than 90% of auxin is in its inactive form [30]. GH3 proteins that correspond to group II catalyze the conjugation of endogenous IAA with various amino acids as the primary mechanism to maintain auxin homeostasis in plants [31]. Some conjugates, such as IAA–Ala, IAA–Leu, and IAA–Phe, can be hydrolyzed to convert IAA back to its free form [8]. IAA–Asp and IAA–Glu conjugates have been thought to serve as intermediates leading to ubiquitin-mediated oxidative degradation of excess IAA in plant tissues [32]. The fate of the conjugates is diverse and is related to the reduction in the endogenous amount of IAA through degradation (IAA-Asp/Glu) or temporary storage in the cell (IAA-Ala/Leu) [28,33]. However, in strawberry, it has been proposed that IAA–Aspp/Glu are hydrolyzed by seedlings, and the IAA is canalized to the signaling pathway [34]. The primary function of the *GH3* family genes is to keep the active IAA’s dynamic homeostasis [35].

Recently, 17 *GH3* family members based on sequences from *A. thaliana*, *Z. mays*, and *O. sativa* have been reported in *C. canephora*, potentially linking putative members GH3 to sequences containing characteristic domains of the GH3 family (PLN02247 and pfam03321). However, the researchers numbered the genes consecutively and did not relate them to the homology of the corresponding members of the *GH3* family in *A. thaliana* [36].

In our laboratory, we have developed an efficient system for obtaining somatic embryos from foliar explants of *C. canephora* [37]. We have used this system to study diverse aspects of the induction of SE [29,38,39,40,41]. Recently, we elaborated a transcriptome of the induction of the SE in *C. canephora* [41]. The transcriptome analysis led to identifying a plethora of *GH3* genes expressed during the induction of SE in *C. canephora*. Here, we present evidence of the participation of some members of the *GH3* family in the induction of SE in *C. canephora*.

## 2. Results

### 2.1. Somatic Embryogenesis Induction in C. canephora

In our laboratory, we have an efficient SE system for *C. canephora* (Figure 1). Our embryogenic model is divided into two stages: pretreatment and induction. In the first stage, the plants are incubated in MS medium supplemented with 1-naphthaleneacetic acid (NAA, 0.54 µM) and kinetin (KIN, 2.32 µM) for 14 days (Figure 1A). At the end of this stage, the second and third pairs of leaves were cut to obtain circular explants (Figure 1B). The circular explants were incubated in a Yasuda medium, with the nitrogen source modified, at the induction stage and supplemented with 6-benzyladenine (BA, 5 µM) [37]. The first embryogenic structures appeared after 21 days in the induction medium (Figure 1C). After 56 days in the induction medium, approximately 300 embryos per explant were obtained at different stages of development (globular, heart, torpedo, and cotyledonary) (Figure 1D,E). The cotyledonar somatic embryos obtained were deposited in MS medium free of plant growth regulators (PGR) for germination. All embryos continued their development until they formed complete seedlings (Figure 1F,G). These plantlets are used to initiate a new cycle of induction of SE.

### 2.2. Ratio of Free and Conjugated Auxin

To have a better understanding of the possible SE activity of *CcGH3* from *C. canephora*, the endogenous concentration of free IAA and some of its conjugates, such as IAA–Ala, IAA–Leu, and IAA–Glu, were quantified by high-resolution liquid chromatography (HPLC) (Figure 2). In foliar explants, IAA’s initial endogenous content was 0.221 nmol g^−1^ FW, which increased significantly during the next 14 days of pretreatment of the seedlings in the presence of NAA and KIN, with a maximum increase of 2.13 nmol g^−1^ FW on day zero. This determination shows that the initial IAA content increased almost 10 times over the initial content (Figure 2).

After induction, the content of free IAA decreased to 0.117 nmol g^−1^ FW during the first 30 min of incubation in the induction medium. Its level increased again after 7 days to 1.27 nmol g^−1^ FW, and the level decreased until day 21 after the induction of SE, precisely when the first pro-embryogenic structures began to appear (Figure 1 and Figure 2). The data on the amount of IAA conjugates present during the SE pre-incubation and induction process also shows an important dynamism and a relationship with the expression of different *CcGH3* genes during the SE process.

We found that the maximum content of the IAA–Glu conjugate was 92.4 nmol g^−1^ FW on day zero; this shows that more than 90% of the auxin was in its conjugated form at this point. The amount decreases drastically during the first 30 min and 1 h of induction in 61.2 nmol g^−1^ FW and 38.3 nmol g^−1^ FW, respectively, practically disappearing after 7 days (Figure 2). The IAA–Ala and IAA–Leu conjugates constituted approximately 10% of the total IAA. The initial content of AIA–Ala was 2.49 nmol g^−1^ FW and increased to 6.33 nmol g^−1^ FW seven days later, reaching a maximum of 11.2 nmol g^−1^ FW on day zero of induction, decreasing during the next 7 and 14 days with 5.12 nmol g^−1^ FW and 2.56 nmol g^−1^ FW, respectively; however, by the time the first embryogenic structures appeared, around day 21 of incubation, the IAA–Ala concentration increased to 9.58 nmol g^−1^ FW (Figure 2).

Regarding the endogenous content of the IAA–Leu conjugate, it seems to have an interesting dynamism. The initial content was 4.43 nmol g^−1^ FW. Then, it decreased dramatically during the first hours of induction; however, 14 days later, it was found to be 4.71 nmol g^−1^ FW, which decreased to 2.32 nmol g-1 FW on day 21 (Figure 2).

### 2.3. Genome-Wide Identification of the GH3 Gene Family in C. canephora

We used the published genome sequence of *C. canephora* [43] to download, visualize, identify and select candidate sequences of the GH3 family using the UGENE software (Appendix A). This family of genes codes for enzymes that catalyze 3-indole-acetic acid conjugation with various amino acids and carbohydrates. Complete information on the *GH3* genes of *C. canephora*, including the gene name, locus ID, homologs in *A. thaliana*, the open reading frame (ORF) length, number of exons, length (aa), molecular mass, isoelectric point of the deduced polypeptide, and subcellular localization, is shown in Table 1.

To identify the possible *GH3* homologs present in *C. canephora*, we used the sequences of the *GH3* genes of *A. thaliana* and used the BLASTP program (https://coffee-genome.org/blast; accessed date 18 May 2020). To avoid any confusion, we used the same nomenclature used in *A. thaliana*. In this way, we identified six *GH3* genes (*CcGH3.1*, *CcGH3.3*, *Ccgh3.5*, *CcGH3.6*, *Ccgh3.9*, and *CcGH3.17*); some of them had more than one copy. For example, four were found for *CcGH3.1*, two for *CcGH3.6* and nine for *CcGH3.17*, giving a total of 18 sequences of the GH3 gene family (Table 1). Most of the identified *CcGH3* genes have similar characteristics among themselves. For example, the open reading frame (ORF) lengths varied between 1284 to 6000 bp, except *CcGH3.3*, which had an ORF of only 862 bp. In addition, each *CcGH3* sequence was aligned with the 19 sequences available for *AtGH3* using the ClustalW program, and phylogenetic trees were constructed independently in the MEGA 7 software.

### 2.4. Phylogenetic Analysis of GH3 Genes in Several Species

The evolutionary difference between monocots and dicots is also reflected in the number of members of each GH3 group. For example, there are 19 *GH3* genes in *A. thaliana* [19], 18 *GH3* genes in *C. canephora* (this study), 13 *GH3* genes in *O. sativa* [20], 15 *GH3* genes in *S. lycopersicum* [22] and eight *GH3* genes in *V. vinifera* [21]. The highest number of members was found in *Brassica napus* (66 members), and when the number of members among the different species is analyzed, we found that group II is the most widely represented (Appendix A). It is also clearly observed that closely related *C. canephora* GH3 proteins (CcGH3-17a to CcGH3.17h) form a phylogenetic clade with *A. thaliana* proteins in group III; however, they are also related to the AtGH3.17 protein of Arabidopsis. By creating a phylogenetic tree representing *A. thaliana* and *C. canephora* combined, we determined a homology between these sequences.

Some members of group II, such as rice, tomato, grape, and possibly coffee, function as IAA synthetases. Some *CcGH3* are grouped with *GH3* from other dicotyledons, such as tomato and *Arabidopsis* in several subgroups (Appendix A). The accession numbers for the related *GH3* genes are listed in Appendix A.

### 2.5. Phylogenetic Analysis of C. canephora and A. thaliana GH3 Genes

The phylogenetic relationship between the *GH3* family members of *C. canephora* and *A. thaliana* was studied by constructing an unrooted phylogenetic tree from the alignments between the 19 *AtGH3* and 18 *CcGH3* sequences. Sequence analysis showed that the genes *CcGH3.1*, *CcGH3.6*, *CcGH3.9*, and *CcGH3.17* are orthologs to the same *AtGH3 A. thaliana*, while *CcGH3.3* and *CcGH3.5* are orthologs to *AtGH3.2* and *AtGH3.11*, respectively. As mentioned before, the *Arabidopsis GH3* can be grouped into three different groups, while in *C. canephora*, most of the *CcGH3* sequences showed higher homology with members of groups I and II of *A. thaliana*. Group I includes only *CcGH3.5* and the two reported genes in *Arabidopsis* (*AtGH3.10* and *AtGH3.11*). In contrast, group II includes the majority of *CcGH3* together with *Arabidopsis* counterparts; interestingly, none revealed homology with members of group III (Figure 3).

### 2.6. Intron–Exon Structure for GH3 Genes in C. canephora

It has been shown that the diversity of the gene structure can play a significant role in the evolution of multigenic families. Therefore, we examined the phylogenetic relationship and exon–intron organization of members of the *CcGH3* family to obtain information on their structural diversity. The results of the analysis showed that the number of exons varies from one to five, while *CcGH3.3* (Cc02_g19470), the smallest member of the family, contains the fewest exons (1), and *CcGH3.17* (Cc10_g16320) contains the highest number of exons (5). Ten members of the *C. canephora GH3* gene family contain four exons, while four members have only three exons. The other two members contain two exons (Figure 4).

### 2.7. Expression Patterns of GH3 Genes Based on RNA-Seq Data and Quantitative Real-Time PCR (qRT-PCR) Analysis

IAA homeostasis is central to the initiation of SE. Without pre-treatment, SE is not obtained [28]. Additionally, during the pre-treatment, there is a very significant increase, both in the concentration of IAA and in some of its conjugates [28]. For this reason, we asked the system if *GH3* was expressed during the induction of SE in *C. canephora*, and if so, which ones. On the other hand, bioinformatic analysis gave us the basis to analyze the presence of the expression of the *GH3* genes in the transcriptome that we previously developed in our laboratory. This transcriptome includes different points of the embryogenic process of *C. canephora* (14 dbi, 9 dbi, 0dbi, 1 dai, 2 dai, and 21 dai) [41]. The *CcGH3* expression profile was analyzed from the RNA-seq data. It was found that 18 *CcGH3* were expressed during at least one point of the embryogenic process (Figure 5). We observed that during the days of pretreatment (14 dbi, 9dbi, and 0 dbi), most of the *CcGH3* had high expression patterns (value > 5), except for *CcGH3.1a* (Cc00_g22520), *CcGH3.6a* (Cc05_g05640), and *CcGH3.17b* (Cc00_g04500), which maintained a low expression throughout the SE process (Figure 5).

To understand how the *CcGH3* genes may be involved during the pretreatment and induction of SE in *C. canephora*, the transcriptome data were validated by qRT-PCR quantitative analysis of the 18 *CcGH3* using specific primers for each gene (Appendix A). These results showed that the expression levels of *CcGH3.1d* (Cc02_g19460), *CcGH3.6b* (Cc05_g12940), and *CcGH3.9* (Cc01_g20620) increased 24h after the induction of SE in an auxin-free medium (Figure 6). It was observed that all *GH3.17* transcripts were present before auxin pretreatment and remained elevated during virtually the entire embryogenic process (Figure 7). On the other hand, *CcGH3.3* (Cc02_g19470) expression increased during the pretreatment period (14 dbi and 9 dbi), just five days after adding exogenous auxin to the culture medium. However, *CcGH3.5* (Cc05_g06700) was only expressed during the first days of the embryogenic process and decreased at the induction stage. This gene is homologous to *AtGH3.11* and is possibly related to the conjugation of JA with isoleucine. 

### 2.8. The Building of 3D Structures, Modelling and Molecular Docking of Selected CcGH3 Proteins

To better understand how CcGH3 catalyzes the synthesis of IAA-amino acid conjugates, we made predictions of three-dimensional (3D) structures for GH3 proteins from *C. canephora*. Those CcGH3 genes that were highly expressed during the SE induction process were selected (Figure 6 and Figure 7). In total, seven *CcGH3* candidates were selected for this study (Figure 8). The amino acid sequences of each CcGH3 protein selected were submitted to SWISS-MODEL to predict their 3D structure by homology (Appendix A). It was determined that all structures of CcGH3 proteins adopt a typical two-domain structure resembling that of adenylating firefly luciferase (ANL) enzyme superfamily proteins: a large N-terminal domain of almost 450 amino acid that corresponds to most of the catalytic site and the rest of the residues of approximately 150 amino acid correspond to a smaller C-terminal domain. The N-terminal domain consists of a fold with two separate β-structures, each surrounded by α-helices, and the C-terminal domain comprises a four-stranded β-sheet flanked by two α-helices (Figure 8A–G).

The multiple alignments of the structures of the CcGH3 proteins selected showed that they share a unique topology (Figure 8H), which suggests that they are structurally conserved. In addition, amino acid sequence alignments of CcGH3 proteins identify conserved sites that could provide substrate specificity through three motifs (Figure 8I).

The 3D structures of the IAA and AMP are showed in Figure 9A,B, respectively. Furthermore, molecular docking showed the shape of AMP and IAA in the large N-terminal domain of the CcGH3 proteins selected in this study (Figure 9C–I). On the other hand, CcGH3 proteins display different preferences both for acidic and non-polar amino acids (Figure 9C–I). The GH3 protein family can enable the binding of IAA to different amino acids, and the particular amino acid used determines the fate of the conjugated auxin: either degradation or storage [44]. In this study, we found that CcGH3.5 has a non-polar amino acid binding motif (Figure 9D), while CcGH3.6a, CcGH3.6b, and CcGH3.9 have acidic amino acid binding motifs (Figure 9E–G), and the motif could not be identified for CcGH3.1b, CcGH3.17c, and CcGH3.17d (Figure 9A,H,I). It is suggested that Asp/Glu are the possible amino acid substrates for most of these CcGH3 proteins. The results showed were determined using the crystallographic data previously reported in AtGH3.5, VvGH3.1, and OsGH3.8 [45,46,47].

## 3. Discussion

With over 2.25 billion cups consumed every day, coffee is one of the most important crops globally. The Coffee Genome Hub [48] is a database that provides centralized access to analysis tools that facilitate basic, translational, and applied research in coffee [43]. In our laboratory, we have worked for several years with tissue culture, particularly to elucidate somatic cells′ embryogenic process. We currently have a system of direct embryogenesis from foliar explants of *C. canephora* Pierre var. Robusta, making it an ideal model for the study of molecular mechanisms that have not responded using zygotic embryogenesis. Although SE has been studied for a long time, the process is not fully understood.

One of the most critical factors during the induction of SE in different species is the addition of PGR to the culture medium [3]. Auxins are the most commonly used PGR and are crucial during the induction of SE [49]. Previous studies have shown that elevated levels of auxin stimulate transcriptional changes related to biological processes such as biosynthesis, conjugation, and transport [50]. For example, during the SE of *C. canephora*, the addition of exogenous auxin induces endogenous IAA accumulation [28]. This increase in the IAA level during pretreatment is due to de novo biosynthesis [29]. Currently, no gene related to the conjugation of IAA with amino acids has been characterized in *C. canephora*.

Pinto et al. [36] recently made a genome-wide analysis and gene expression profile of GH3 genes in *Coffea* spp. However, the main differences that can be highlighted between their study and ours are the number of identified and quantified GH3 genes, the biological model, the type of explants used, and the quantification of auxin. Pinto et al. [36] made an analysis of the GH3 genes present in the genome of *C. canephora* and evaluated the expression profiles of four GH3 genes in non-embryogenic cells, embryogenic cell suspension, and embryogenic cells of *C. arabica*; no quantification of free and conjugated auxin was performed. In our study, we identified and evaluated the expression of the 18 GH3 genes directly throughout our SE induction process; at the same time, the free and conjugated auxin was quantified. With the above, we concluded that GH3 could be regulating the induction of SE in *C. canephora* through the conjugation of auxin with acidic amino acids as a substrate. It should also be noted that *C. canephora* is diploid, while *C. arabica* is tetraploid, the product of the cross between *C. canephora* and *C. eugenioides*. Regarding the ploidy, what is known so far is that different numbers of ploidy influence the embryogenic response of the different species belonging to the genus *Coffea*. The average number of cotyledonary somatic embryos was different according to the level of ploidy of the species, even under homogeneous culture conditions during the induction of indirect SE [51]. The induction of SE is influenced by epigenetic changes, the conditions of culture, the medium′s composition, and the type of explant used. In order to make a fair comparison between the ploidy level and SE, a thorough investigation of DNA methylation is recommended, as it is known that methylation has a role in gene regulation and consequently in plant growth and development processes [51,52].

In general, the distribution of the *C. canephora* gene family is consistent compared to the other gene families in plants [48]. In this study, we identified 18 *CcGH3* genes named according to their homology with *A. thaliana GH3* genes (Table 1). We also built a phylogenetic tree to analyze the relationship of *GH3* gene families between *Coffea* and *Arabidopsis*. We found genes with high starting values (≥95%), suggesting that *CcGH3* was very homologous with *Arabidopsis* (Figure 3). The classification of gene families in the genome of *C. canephora* is based on sequences of grape, tomato, and *Arabidopsis* proteins [43]. On the other hand, *C. canephora* and *S. lycopersicum* share a common ancestor. This may suggest that duplication events gave rise to these genes and occurred before the divergence of some Solanaceae from other dicot families. These phylogenetic relationships suggest a possible evolutionarily conserved function of GH3 proteins in the regulation of auxin homeostasis.

Previously, *Arabidopsis* GH3 proteins have been classified into three groups (I, II, and III) according to their sequence homology based on phylogenetic analyses, on the specificity to adenylate plant hormones, and on the study of mutants with loss or gain of function (Appendix A) [16,53]. Although the GH3 family is highly conserved in several species, a study based on *EST* sequences of different monocots, including *Z. mays*, *Triticum aestivum*, *Hordeum vulgare*, *Sorghum bicolor*, and *O. sativa* [54], suggests that group III of GH3 is absent in monocots [20].

In the phylogenetic tree, we observed that *CcGH3.5* is grouped only with *AtGH3.10* and *AtGH3.11*, suggesting that they possibly fulfill a similar function. It has been reported that *AtGH3.11/JAR1* is the only enzyme in group I of *A. thaliana* that functions as JA-amino synthetase [55]. This provides a mechanism for JA regulation by conjugation with isoleucine, which leads to the formation of bioactive jasmonate (JA-Ile) [53]. Meanwhile, *AtGH3.10* is related to the mutant *AtGH3.10/dfl2-D* (*dwarf in light 2*) involved in the elongation of the hypocotyl in the presence of red light; however, enzymatic activity for the protein has not yet been reported [19].

On the other hand, in the phylogenetic tree, it was observed that the rest of the *CcGH3* are included within group II of *AtGH3* and possibly act as IAA–amide synthetases. Interestingly, group III enzymes have only been identified in *A. thaliana*. However, so far, their function has not been determined, and they appear not to be active in the presence of IAA or other PGR [15].

The relative expression of most *CcGH3* during pretreatment and induction of SE in *C. canephora* suggests the participation of IAA-amido synthetases in regulating the endogenous content of IAA to cope with excess auxin by conjugation with amino acids. In *C. canephora*, it has been reported that the increase in *CcGH3.1* and *CcGH3.17* transcripts correlates with the increase in conjugated IAA [28]. On the other hand, in *V. vinifera*, it was determined that the increase in the transcription of *VvGH3.1* is related to the establishment and maintenance of low IAA concentrations [56].

Most of the results for qRT-PCR analysis are similar to those found in our transcriptome′s expression profile. These data show the close relationship between both experiments and, at the same time, validate the transcriptome data (Figure 5). We suggest that some differences may be related to the biological variation between the different samples used for each experiment. These data suggest that *CcGH3* plays a vital role during the preconditioning and induction stage of SE.

One of the most notable results when comparing the free and conjugated auxin quantification data and the relative expression analysis is that one day after the start of the induction stage, there is a decrease in the amount of free auxin. At the same time, the amount of IAA–Ala and IAA–Leu conjugates increases. At the same points, the relative expression of specific GH3 genes is also increased. This increase suggests a regulation of auxin concentration through conjugation to give rise to SE.

In *Arabidopsis*, at least seven *AtGH3* belonging to group II are induced by auxins and function as IAA–amide synthetases (Appendix A). It has been reported that *AtGH3.5*, *AtGH3.6*, and *AtGH3.17* are related to the conjugation of auxin with different amino acids (such as aspartic, glutamic, alanine, leucine, and tryptophan) adjusting the levels of active auxin inside the cell, providing a regulation mechanism [16,57]. In *C. canephora*, *CcGH3.6a*, and *CcGH3.6b* (Cc05_g05640 and Cc05_g12940), homologous to *AtGH3.6*, were highly expressed one day after the induction. This increase in the expression of these genes correlates with the increase in the amount of IAA–Glu (Figure 2). All *CcGH3.17* were highly induced and are expressed throughout the SE process In *A. thaliana*, the transcription of *AtGH3.17* is accompanied by the increase in IAA–Glu and IAA–Asp conjugate, related to the degradation of excess auxin [16].

In *Arabidopsis*, when auxin homeostasis and the average level of *GH3* transcripts are altered, different phenotypes are observed [24]. Most relevant studies have used mutants that overexpress *GH3* in different species. For example, in rice, the overexpression of *OsGH3.8* resulted in plants with slow growth, atypical morphology, and high levels of IAA-Asp conjugate [58]. Additionally, the overexpression of *OsGH3.13*, an ortholog of *GH3.17* from *A. thaliana*, results in alterations in the plant′s architecture, showing a dwarf deficient phenotype of auxin, as well as an increase in the conjugates IAA–Asp and IAA–Glu [59]. In *A. thaliana*, *AtGH3.6/dfl1-D*, an auxin-response *GH3* gene homolog, negatively regulates the lengthening of hypocotyl and lateral roots, resulting in dwarf phenotypes [60].

Auxin controls different cellular processes, including the induction of somatic embryogenesis, so its levels must be precisely regulated. Auxin homeostasis can be regulated in various ways, and endogenous levels can be altered by its conjugation [29,61]. There is increasing evidence that conjugation reactions through GH3 play an important role in these regulatory processes [24], in particular, the inactivation of IAA through conjugation with amino acids [16].

Little is known about the kinetic and chemical mechanisms of the GH3 enzymes. It has been reported that GH3 functions as an IAA–amido synthetase [57], which catalyzes the ATP-dependent formation of IAA–amino acid conjugates [47]. In conjugation reactions by the GH3 family, ATP binds first, followed by IAA. Then, the formation of adenylated IAA (acyl-AMP) intermediate results in the release of pyrophosphate. Subsequently, the activated acyl-AMP intermediate undergoes a nucleophilic attack from the amino acid to yield IAA–amino acid conjugate, releasing AMP [45,47].

The GH3 proteins analyzed have a large N-terminal domain and a small C-terminal domain, and the active site is located between these two domains. Nucleotide-binding sites contain three conserved motifs; motifs are found close to the active site and play a key role in efficient catalysis. The overall architecture of the CcGH3 structures predicted in this study was similar to the OsGH3-8 structure reported recently [47]. This study also reported the identification of a conserved region that could function in nucleotide binding through three motifs in three plant species: *O. sativa*, *A. thaliana*, and *V. vinifera* [47].

Until now, *GH3* gene participation has been reported mainly with mutants with loss or gain of function. However, a more in-depth analysis of the structure and function is needed to have a more precise understanding of how these GH3 proteins regulate auxin homeostasis. Our data suggest that most of the *CcGH3* genes′ expression presents changes that correlate with the formation of the first embryogenic structures. These results suggest that the addition of exogenous auxin stimulates changes in the transcription of auxin-sensitive genes, which could be involved in homeostasis and during the induction of the embryogenic process of *C. canephora*.

## 4. Materials and Methods

### 4.1. Biological Material and Growth Conditions

Plantlets of *C. canephora* Pierre var. Robusta were grown under in vitro conditions in a semi-solid maintenance Murashige-Skoog (MS) [62] medium supplemented with 29.6 µM thiamine-HCl (Sigma T3902), 550 µM myo-inositol (Sigma, I5125), 0.15 µM L-cysteine (Sigma, C-8277), 87.64 mM sucrose (Sigma, S539), and 0.285% (*w*/*v*) Gellan gum (PhytoTechnology Laboratories, G434), under photoperiod conditions of 16 h light/8 h dark (150 µmol m^−2^ s^−1^) at 25 ± 2 °C. The medium was adjusted to pH 5.8 and sterilized at 121 °C (1 kg cm^−2^) for 20 min.

### 4.2. Somatic Embryogenesis Induction

The SE induction process encompasses two stages: pretreatment and induction (Appendix A). First, the seedlings were preconditioned for 14 days in a semi-solid MS [62] medium supplemented with 0.54 µM NAA (Sigma, N-1145) and 2.32 µM KIN (Sigma, K0753) under the same conditions as described above. At the end of the pretreatment, the second and third pair of leaves were selected and cut into segments of 0.8 cm in diameter with the help of a sterile punch. Five explants were placed per 250 mL flask, with 50 mL of Yasuda liquid medium with the nitrogen source modified [42] and supplemented with 5 µM of BA (PhytoTechnology Laboratories, B800), adjusted to pH 5.8 [37]. The flasks were incubated in the dark and shaken (55 rpm) at 25 ± 2 °C for 56 days.

### 4.3. Auxins and Auxin Conjugates Extraction

One hundred milligrams of plantlet tissue was collected from the beginning of preincubation (days-14 and -7 before induction) to the induction day (day zero). Samples were also collected at 0.02, 0.04, 1, 7, 14, and 21 days after SE induction. The collected tissue was frozen and stored until its use. All the analyses were performed with three biological replicates from at least two independent experiments.

The frozen tissue was ground with liquid nitrogen and mixed with 1 mL of acidic water (the pH was adjusted to 2.8 with HCl). The mixture was transferred to a test tube with an additional milliliter of acidic water. The mixture was stirred for 1 min with 1 mL of a solution of butylated hydroxytoluene (Acros Organics, 112992500, Thermo Fisher Scientific), and then 1 mL of ethyl acetate (CTR Scientific 00184) was added. The mixture was stirred for 1 min, and the supernatant recovered. Next, 2 mL of ethyl acetate was added, stirred for 1 min, and recovered the supernatant. This operation was repeated once more. From this mixture, 3 mL of the organic phase was taken and evaporated with nitrogen gas. The dried sample was resuspended in 1 mL of the mobile phase and filtered through a Millipore filter (0.22 µM). The preparation of IAA–Glu, IAA–Leu, and IAA–Asp was previously reported [28].

### 4.4. High-Performance Liquid Chromatography

For the analysis of the samples, an Agilent Technologies 1200 high-resolution liquid chromatography (HPLC) system consisting of a quaternary array of pumps (Agilent Technologies G1311A) connected to an automatic injector (Agilent Technologies G1329A) was used. A total of 20 μL of the tissue extract was injected and subjected to chromatography with an isocratic elution system with a flow rate of 0.6 mL min^−1^ in a C_18_ reverse-phase column (Phenomenex, Torrance) of 250 mm × 4.6 mm. The samples were analyzed with a fluorescence detector (Agilent Technologies G1321A) at an emission length of 280 nm and an excitation length of 340 nm.

### 4.5. RNA Extraction and cDNA Synthesis

Total RNA was isolated from 50 mg of leaf tissue (between five and seven circular explants) using the Direct-zol RNA MiniPrep Kit (Zymo Research, R2051) according to the manufacturer’s instructions. To avoid degradation, all steps were carried out at low temperature. The quantity and quality of RNA was determined using the NanoDrop^TM^ 2000 spectrophotometer (Thermo Scientific) and gel electrophoresis. Approximately one µg of qualified total RNA of each sample was used to synthesize first strand complementary DNA according to the manufacturer’s instructions of the RevertAid H Minus First Strand cDNA Synthesis Kit (Thermo Scientific, K1632).

### 4.6. Quantitative Real-Time PCR (qRT-PCR) Analysis

Due to the availability of the *C. canephora* genome [43], specific primers were designed for each gene of interest. Primers for qRT-PCR were designed using the NCBI Primer-Blast Tool (https://www.ncbi.nlm.nih.gov/tools/primer-blast/; accessed date 10 June 2020), taking into account several parameters in the design. The length of the primers was established between 20 and 30 bp, a GC content between 45 and 60% was considered, as was a melting temperature between 59 °C and 60 °C. The amplification ranges of the amplicon were established between 150 and 210 bp. In addition, the probability of secondary structure formation and dimerization was verified in the Oligo Explorer 1.4 software (http://www.genelink.com/tools/gl-oe.asp; accessed date 10 June 2020). Detailed information about all the primers used in this study is listed in Appendix A. The fluorescence and cycle values provided by the real-time PCR equipment were exported in an excel sheet. These data were then imported into the LinRegPCR version 2021.1 software to verify that the estimate of the efficiency of the primers was greater than 90% [63,64]. Each reaction contained 100 ng of cDNA template, 10 µM of each primer and 1x Express Sybr Green^®^ ER qPCR Supermix Universal (Invitrogen, A10314). The real-time PCR analysis was performed on a StepOne Real-Time PCR System (Applied Biosystems) under the following conditions: 1 cycle of 95 °C 5 min, followed by 40 cycles of 95 °C 15 s and 55 °C 30 s. Data obtained from real-time PCR were used to calculate the relative expression levels of the target gene and compared with the expression of the cyclophilin gene using the 2^−ΔΔCT^ method [65]. The experiment was conducted with three biological replicates, using the cyclophilin gene as an internal reference [41,66].

### 4.7. Genome- and Transcriptome-Wide Identification of GH3 Genes in C. canephora

To carry out the identification of the sequences of the *GH3* genes, a search was carried out using the keyword “GH3” within the reference genome of *C. canephora* available online (http://coffee-genome.org/; accessed date 18 May 2020) [48]. A total of 20 candidate sequences were found. Since the *GH3* family consists of characteristic domains, all recovered GH3 sequences were analyzed in the NCBI conserved domain database (https://www.ncbi.nlm.nih.gov/Structure/cdd/wrpsb.cgi; accessed date 22 June 2020). Eighteen sequences coincided with the *GH3* domain (pfam03321 and PLN2247) considered in candidate GH3 sequences. For the nomenclature of *GH3* in *C. canephora*, 19 GH3 sequences (GH3.1 to GH3.19) were downloaded from The Arabidopsis Information Resource database (TAIR). A blast analysis was subsequently performed against the GH3 sequences of *C. canephora*.

A manual search using the keyword “GH3” was performed within the transcriptome data of our SE induction process [41]. RNA-seq data are available at NCBI GEO: GSE128888. After identifying the GH3 genes, a heatmap was generated showing the expression levels, using the ggplot2 package for R [67].

### 4.8. Phylogenetic Analysis

The *GH3* sequences were aligned with ClustalW. Subsequently, a phylogenetic tree was built using the MEGA 7 software (http://www.megasoftware.net/; 15 July 2021). The evolutionary history was inferred using the maximum likelihood method based on the general time reversible model. The tree with the highest log likelihood (–9815.16) is shown. Initial tree(s) for the heuristic search were obtained by applying the neighbor-joining method to a matrix of pairwise distances estimated using the maximum composite likelihood (MCL) approach. A discrete gamma distribution was used to model evolutionary rate differences among sites (two categories (+*G*, parameter = 0.7788)). The analysis involved 72 sequences. All positions with less than 95% site coverage were eliminated. Less than 5% alignment gaps, missing data, and ambiguous bases were allowed at any position. There were a total of 700 positions in the final dataset. The percentage of replicate trees in which the associated taxa clustered together in the bootstrap test (1000 replicates) is shown next to the branches. The sequences of *C. canephora GH3* genes were obtained from http://coffee-genome.org. Rice sequences were obtained from http://riceplantbiology.msu.edu; accessed date 18 May 2020. Tomato sequences were obtained from https://solgenomics.net/; accessed date 18 May 2020. *Arabidopsis* sequences were obtained from https://www.arabidopsis.org/; accessed date 19 May 2020. Grape sequences were obtained from NCBI https://www.ncbi.nlm.nih.gov/; accessed date 19 May 2020.

### 4.9. Systematic Bioinformatics Analysis of C. canephora GH3 Genes

For genetic structure analysis, the complete annotation of *C. canephora* was downloaded directly from the genome database (http://coffee-genome.org/; accessed date 18 May 2020) and the sequences were visualized with Unipro UGENE online software (http://ugene.net/; accessed date 22 June 2020). The online software PIECE (Plant Intron Exon Comparison and Evolution database; https://wheat.pw.usda.gov/piece/index.php/; accessed date 28 July 2020) was used to predict the intron–exon organization. The length, molecular mass, and theoretical isoelectric point of each CcGH3 protein were calculated with the software ExPASy (https://web.expasy-org/protparam/; accessed date 29 July 2020) using the input of coding sequences (CDSs) and corresponding genomic sequences. The subcellular localization was predicted on the PSORTII (https://psort.hgc.jp/form2.html//; accessed date 31 July 2020). 

### 4.10. The Building of 3D Structures, Modelling, and Molecular Docking of Selected CcGH3 Proteins in C. canephora

All 3D structures were built using the SWISS-MODEL software, accessible via the ExPASy web server (https://swissmodel.expasy.org//; accessed date 17 February 2021). The best-predicted models were evaluated using global model quality estimation (GMQE) and assessed after model building using the QMEAN global score. The Chimera MatchMaker tool [68] was used to compare the homology modeling structure of the nine CcGH3 proteins. UCSF Chimera 1.14 software was used to model and visualize the 3D structures [68].

Molecular docking experiments were carried out to evaluate the binding sites of AMP and IAA to the predicted CcGH3 structural model, using DockThor server (https://dockthor.lncc.br/v2/; accessed date 12 April 2021). The DockThor scoring function is based on the MMFF94S force field [69]. A standard docking mode was performed using the 3D structures of AMP and IAA. The 3D AMP and IAA structures were built from the molecular formula using the structure edition tool with the build structure option of the UCSF Chimera 1.14 software. The molecular formulas were downloaded from PubChem https://pubchem.ncbi.nlm.nih.gov//; accessed date 15 April 2021. A grid box with blind molecular docking was used to define the docking region. The parameters are referred to as defaults in DockThor, and the structures with positional root mean square deviation (RMSD) o up to 3 Å were clustered together. All results were analyzed using UCSF Chimera 1.14 Molecular Graphics Systems [68].

### 4.11. Statistical Analysis

The data were processed and analyzed using an analysis of variance (ANOVA) program. The significance grade between the mean values was determined using the Tukey test. Differences were considered to be significant at *p* ≤ 0.05. Data were analyzed by Origin 8 (Data Analysis and Graphing Software).

## 5. Conclusions

There are several plant development processes related to auxin signaling. The understanding of the biological functions of *GH3* genes has advanced significantly in recent years. However, the possible function of each of these genes may vary between different plant species. In *C. canephora*, an expression analysis of the SE induction process had not been reported. The phylogenetic evidence of GH3 proteins shows two main groups that correspond to the functional groups identified in *A. thaliana*, suggesting similar functions in dicotyledons. Within the genome of *C. canephora*, 18 *CcGH3* genes were identified and expressed during at least one point of the SE process. Quantitative analysis by qRT-PCR revealed that *CcGH3.6* and *CcGH3.17* are expressed mostly 24 h after SE induction. The analysis of the CcGH3 proteins shown in this work indicates that they can influence the induction of somatic embryogenesis of *C. canephora* through the conjugation of auxin with acidic amino acids as a substrate, which leads to the degradation of excess auxin. However, CcGH3 proteins can also conjugate other growth regulators, such as JA and SA. In plants, SA mediates disease resistance and abiotic stress responses. All these results suggest that the coffee GH3 genes may influence the homeostasis of the auxin content in the cell, necessary to change its genetic program to become a new organism. In general, the data presented in this study on CcGH3 proteins provide information to understand the basic mechanism of somatic embryogenesis in non-model species.

The results presented in this study provide information for understanding the molecular mechanisms of the embryogenic process. They can serve as a basis for the functional characterization of members of the *GH3* family in *C. canephora*. The analysis of the subcellular location and the functional characterization of these genes could help reveal the control mechanisms of the pathway related to auxin conjugation. On the other hand, there is the possibility that some *GH3* act redundantly. Genetic editing tools, such as CRISPR/Cas9, could provide a valuable means to explore the specific function of genes involved in SE.

## Figures and Tables

**Figure 1 plants-10-02034-f001:**
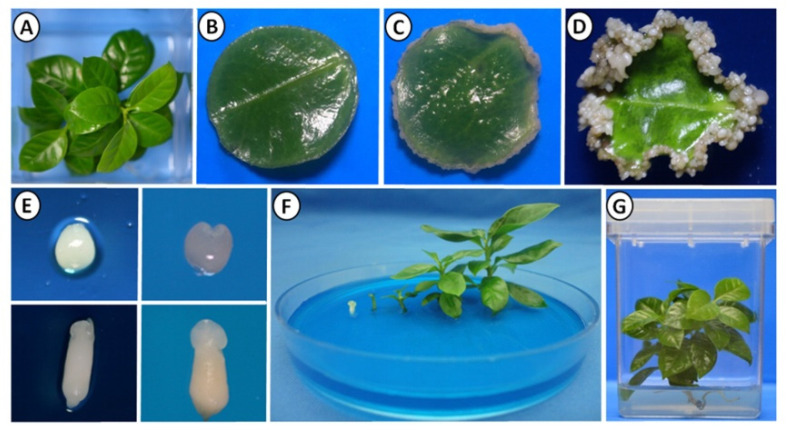
Induction of somatic embryogenesis in *C. canephora*. The SE process was induced, as previously reported. (**A**) Plantlet in the pre-conditioning stage, in MS medium, supplemented with 0.54 μM NAA and 2.32 μM KIN. (**B**) Explant at seven days after SE induction (dai) in Yasuda medium [42] supplemented with 5 μM BA. (**C**) Explant at 21 days after SE induction (dai). (**D**) Explant at 56 days after SE induction. (**E**) Embryos at different stages of development (globular, heart, torpedo, cotyledonary). (**F**) Embryo germination in MS medium without PGR. (**G**) Complete plantlets regenerated from somatic embryos.

**Figure 2 plants-10-02034-f002:**
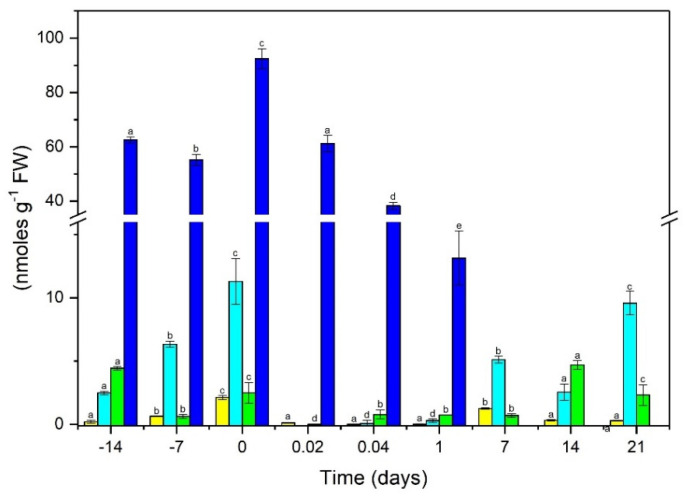
Endogenous IAA and IAA conjugate content, before and during the induction of somatic embryogenesis in *C. canephora*. Endogenous free IAA (yellow bars), IAA–Ala (Cyan bars), IAA–Leu (green bars), IAA–Glu (blue bars). One hundred mg of tissue was collected from the beginning of the seedlings′ preincubation (days-14, -7, and -4) until the induction day (day zero). Samples 0.02, 0.04, 1, 7, 14, and 21 days after induction of SE were also collected. The samples were analyzed as described in Materials and Methods. All analyses were performed with three biological replicates of at least two independent experiments. The error bars represent the standard error (n = 3). Different letters represent the statistical significance of mean differences between each determination at a given time according to the Tukey test (*p* ≤ 0.05).

**Figure 3 plants-10-02034-f003:**
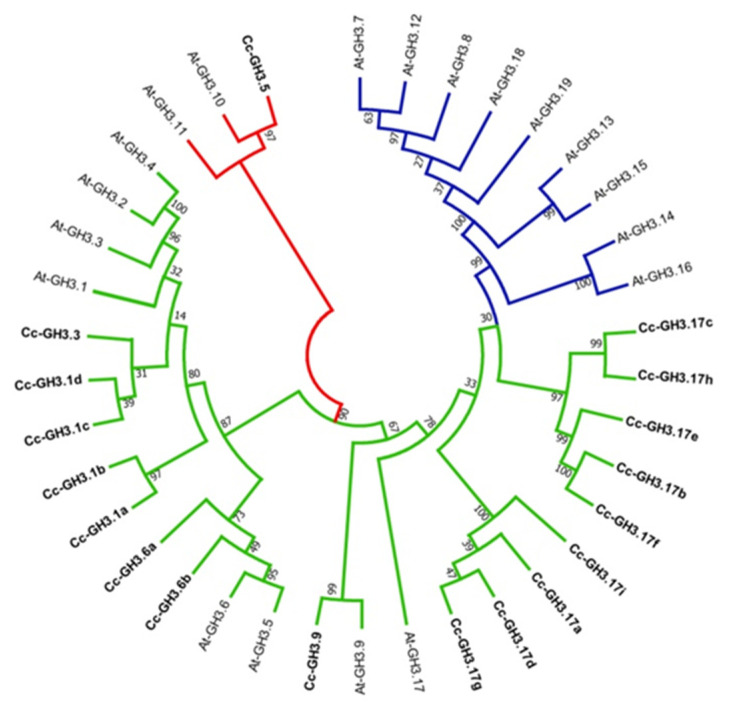
Analysis of the phylogenetic relationships between *CcGH3* and *AtGH3* members. The non-rooted phylogenetic tree was constructed to study the phylogenetic relationship from the sequence alignments of 19 *AtGH3* and 18 *CcGH3*. GH3s belonging to one plant species are marked with the indicated leaf labels. The branches in red, green, and blue colors represent the groups I, II, and III, respectively. The sequences were aligned in ClustalW, and analysis was conducted in MEGA7 using the neighbor-joining method. Bootstrap values (1000) are presented for all branches. At: *A. thaliana*; Cc: *C. canephora*.

**Figure 4 plants-10-02034-f004:**
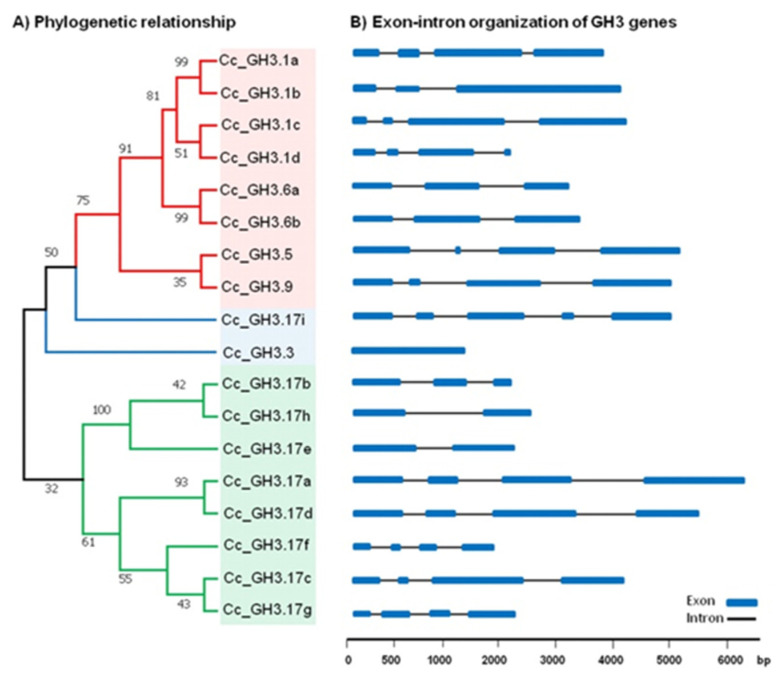
Phylogenetic relationship and intron–exon structure of the 18 *GH3* genes of *C. canephora*. (**A**) The phylogenetic tree was constructed via the alignment of full-length amino acid sequences from *C. canephora* using MEGA 7 software and the neighbor-joining method. The tree clearly exhibits four subgroups. (**B**) The exon and intron lengths of each *GH3* gene are shown proportionally and were created using the genetic structure drawing server found in the PIECE database. Most of the genes have between three to five exons, while *CcGH3.3* (Cc02_g19470) has only one. The blue squares and black lines represent exons and introns, respectively.

**Figure 5 plants-10-02034-f005:**
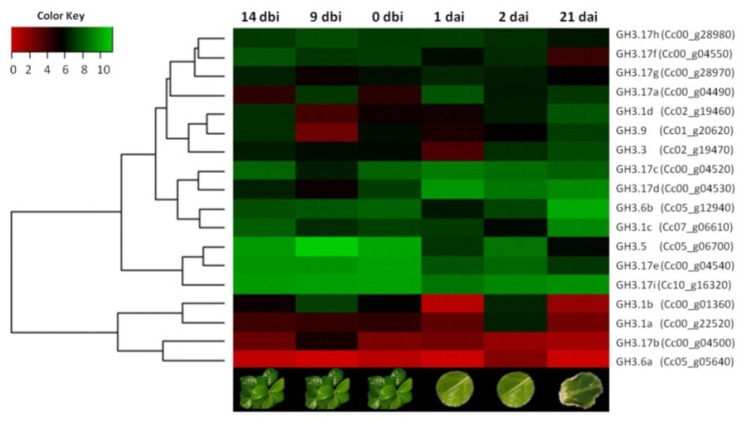
Heat map of expression profiles of the *C. canephora GH3* gene family during the induction of SE. Red, black and green backgrounds represent low, intermediate, and high expression levels, respectively, and are shown on a log2 scale from the highest to the lowest expression of each *CcGH3*. The expression data values were median-centered and normalized for each gene at different points before transforming to the color scale. The abbreviations “dbi” and “dai” indicate the days before induction and days after induction.

**Figure 6 plants-10-02034-f006:**
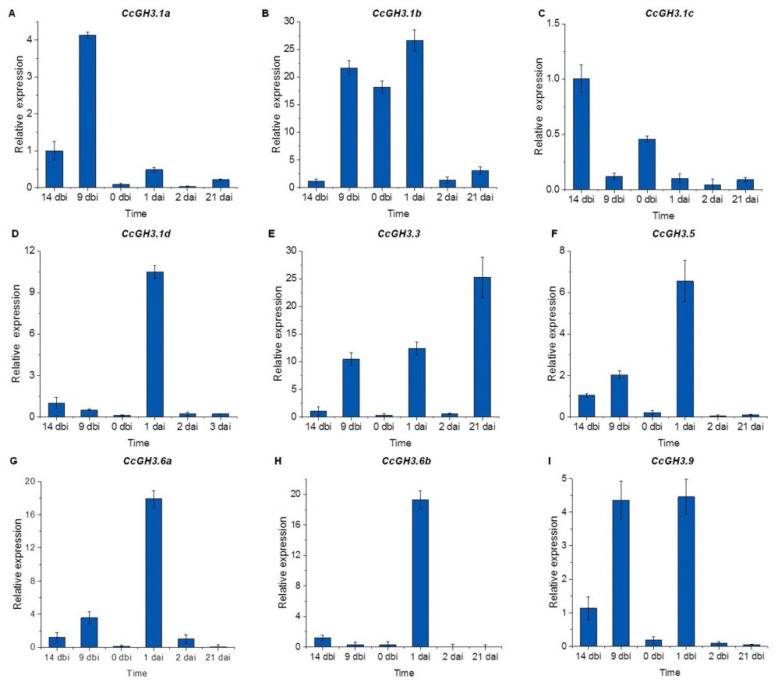
Analysis of the expression of the genes *CcGH3* (*CcGH3.1a* to *CcGH3.9*) during the SE process in *C. canephora*. The expression level of individual *GH3* genes is shown. Fifty mg of tissue was collected from the beginning of the preincubation of the plantlets (days-14 and -9) to the induction day (day zero). Samples were also collected 1, 2, and 21 d after the induction of SE. Samples were analyzed as described in Materials and Methods. Error bars represent the standard error (n = 3).

**Figure 7 plants-10-02034-f007:**
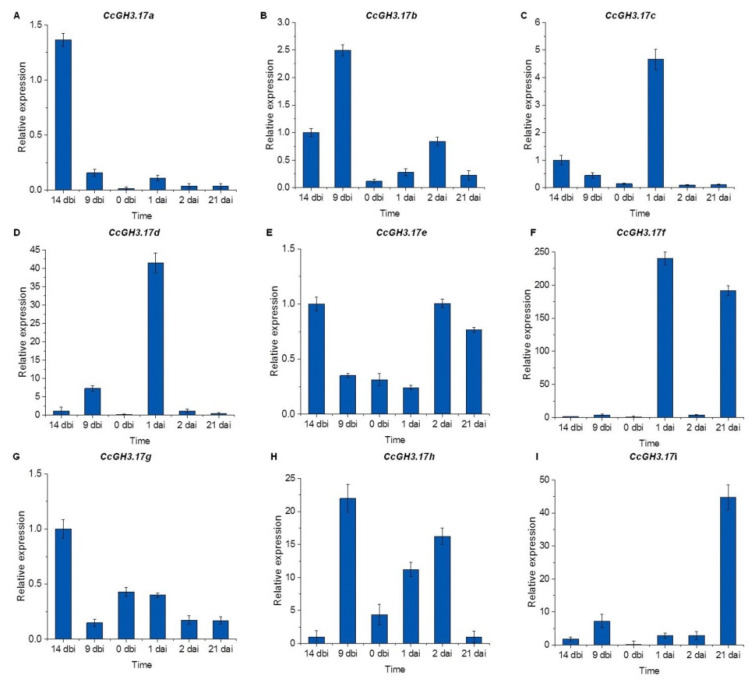
Analysis of the expression of the genes *CcGH3* (*CcGH3.17a* to *CcGH3.17i*) during the SE process in *C. canephora*. The expression level of individual *GH3* genes is shown. Fifty mg of tissue was collected from the beginning of the preincubation of the plantlets (days-14 and -9) to the induction day (day zero). Samples were also collected 1, 2, and 21 d after the induction of SE. Samples were analyzed as described in Materials and Methods. Error bars represent the standard error (n = 3).

**Figure 8 plants-10-02034-f008:**
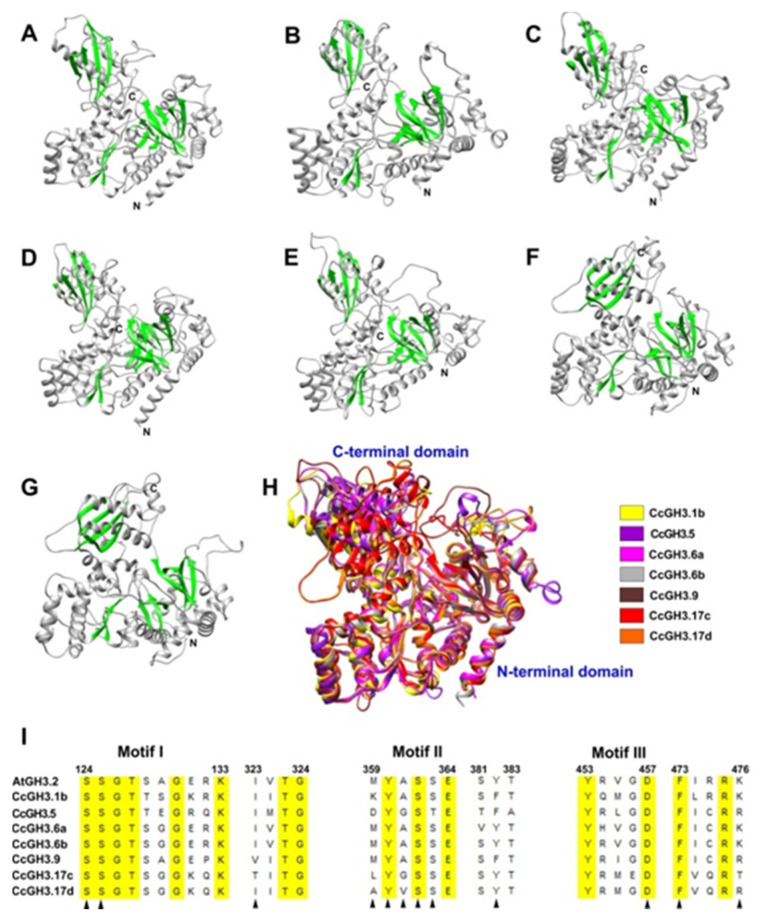
Predicted 3D structures of CcGH3 proteins in *C. canephora*. Panel (**A**): CcGH3.1; panel (**B**): CcGH3.5; panel (**C**): CcGH3.6a; panel (**D**): CcGH3.6b; panel (**E**): CcGH3.9; panel (**F**): CcGH3.17c; panel (**G**): CcGH3.17d; panel (**H**): multiple alignments of selected CcGH3 proteins. The α helices are shown in green, the β strands and random coils in gray. Panel (**I**): sequence alignment of CcGH3 selected with AtGH3.2 of the *A. thaliana*. Highly conserved residues are shown with a yellow background. Black triangles denote the residues involved in AMP binding.

**Figure 9 plants-10-02034-f009:**
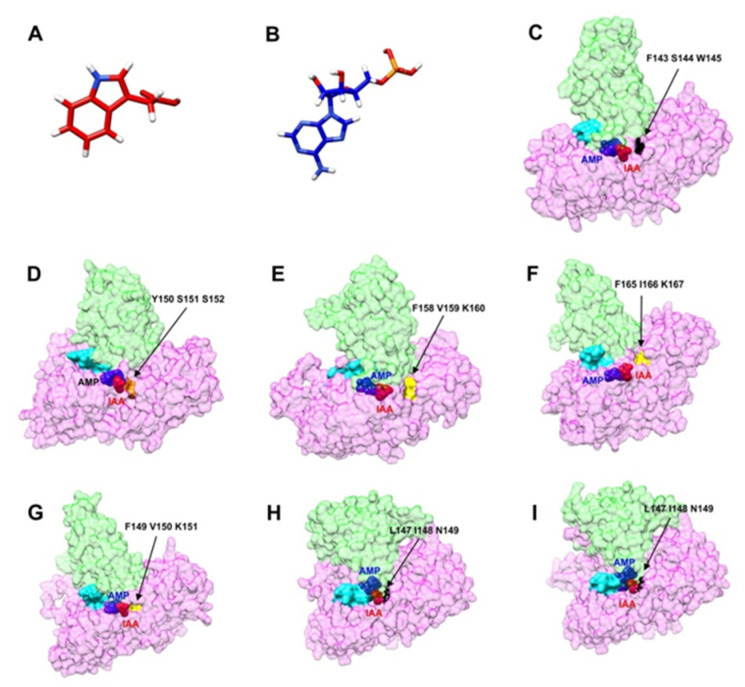
CcGH3-IAA-AMP complexes. Panel (**A**,**B**) are the 3D structures of the IAA and AMP, respectively. (**C**–**I**), surface view of IAA (red spheres) and AMP (blue spheres) binding site in CcGH3 in *C. canephora*. IAA and AMP in the active site are shown as spheres in CcGH3 proteins. The colors green and magenta indicate the small C-terminal domain and the large N-terminal domain, respectively. Arrows show the amino acid binding site: cyan, a motif I of binding to AMP; orange, non-polar amino acid binding site; yellow, acidic amino acid binding site; areas in in black were not identified. Panel (**C**): CcGH3.1; panel (**D**): CcGH3.5; panel (**E**): CcGH3.6a; panel (**F**): CcGH3.6b; panel (**G**): CcGH3.9; panel (**H**): CcGH3.17c; panel (**I**): CcGH3.17d.

**Table 1 plants-10-02034-t001:** Characteristics of the *GH3* gene family members in *C. canephora*.

Gene	Locus ID	ORF (bp)	TAIR Locus ID	No. Exons	Start	End	Deduced Polypeptide			Predicted Subcellular Localization
							Length (aa)	MW (Da)	pI	
*CcGH3.1a*	Cc00_g22520	2581	AT2G14960	4	142,656,917	142,659,058	569	64,641.15	5.69	Cytoplasmic
*CcGH3.1b*	Cc00_g01360	2740	AT2G14960	3	8,822,291	8,824,450	593	66,898.26	5.35	Cytoplasmic
*CcGH3.1c*	Cc07_g06610	2750	AT2G14960	4	4,821,858	4,824,041	528	59,892.74	6.50	Cytoplasmic
*CcGH3.1d*	Cc02_g19460	1367	AT2G14960	4	17,549,243	17,550,429	271	30,560.21	8.26	Cytoplasmic
*CcGH3.3*	Cc02_g19470	862	AT4G37390*AtGH3.2/YDK1*	1	17,550,591	17,551,298	236	26,644.15	5.61	Cytoplasmic
*CcGH3.5*	Cc05_g06700	3771	AT2G46370*AtGH3.11/JAR1*	4	21,465,217	21,468,524	591	67,271.46	5.91	Cytoplasmic
*CcGH3.6a*	Cc05_g05640	2379	AT5G54510*AtGH3.6/DFL1*	3	20,228,391	20,230,769	607	68,277.83	5.53	Cytoplasmic
*CcGH3.6b*	Cc05_g12940	2430	AT5G54510*AtGH3.6/DFL1*	3	26,669,847	26,672,091	622	69,873.15	6.09	Cytoplasmic
*CcGH3.9*	Cc01_g20620	2990	AT2G47750*AtGH3.9*	4	37,172,864	37,175,503	606	68,693.72	5.93	Cytoplasmic
*CcGH3.17a*	Cc00_g04490	6755	AT1G28130*AtGH3.17*	4	34,209,474	34,211,882	583	65,979.47	5.66	Cytoplasmic
*CcGH3.17b*	Cc00_g04500	1516	AT1G28130*AtGH3.17*	3	34,230,766	34,232,281	371	42,536.80	6.19	---
*CcGH3.17c*	Cc00_g04520	2705	AT1G28130*AtGH3.17*	4	34,265,456	34,267,828	583	65,990.42	5.77	E. reticulum
*CcGH3.17d*	Cc00_g04530	4103	AT1G28130*AtGH3.17*	4	34,279,931	34,282,340	583	66,020.49	5.73	Cytoplasmic
*CcGH3.17e*	Cc00_g04540	1774	AT1G28130*AtGH3.17*	2	34,297,176	34,298,627	357	40,498.45	5.69	Cytoplasmic
*CcGH3.17f*	Cc00_g04550	1284	AT1G28130*AtGH3.17*	4	34,298,709	34,299,719	226	25,308.99	5.56	Cytoplasmic
*CcGH3.17g*	Cc00_g28970	1939	AT1G28130*AtGH3.17*	4	178,935,475	178,936,518	239	26,550.34	6.44	Cytoplasmic
*CcGH3.17h*	Cc00_g28980	1961	AT1G28130*AtGH3.17*	2	178,936,581	178,938,006	348	39,719.58	5.78	Cytoplasmic
*CcGH3.17i*	Cc10_g16320	3565	AT1G28130*AtGH3.17*	5	27,266,812	27,269,856	583	65,421.04	5.44	Cytoplasmic

## Data Availability

No applicable.

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
