# Peer review of "Genome-Wide Analysis, Modeling, and Identification of Amino Acid Binding Motifs Suggest the Involvement of GH3 Genes during Somatic Embryogenesis of Coffea canephora"

_plants, 2021, doi:10.3390/plants10102034_

Round 1
Reviewer 1 Report
plants-1323207-peer-review-v1
Full Title: Genome-wide analysis, modeling and identification of amino acid binding motifs suggest the involvement of GH3 genes during somatic embryogenesis of Coffea canéfora
General comments to the manuscript:
The manuscript presented by Méndez-Hernández and co-authors deals with a study that aims to provide further knowledge about the role of CH3 genes on somatic embryogenesis (SE) in Coffea canephora. Besides the characterization of gene family composition, authors also try to establish a link between CH3 transcript level and auxin metabolism during SE.
The topic of the manuscript is interesting and reaches the interest of Plants’ readers. However, I cannot identify the novelty of the research here presented in comparison to the manuscript published by Pinto et al. (2019): https://doi.org/10.1186/s12864-019-6176-1. The authors included the analysis of endogenous IAA and IAA conjugate content but the remaining analysis are highly similar to previous research. I can understand that authors want to rename the CH3 genes based on Arabidopsis members (page 3, line 100) but the reason why they want to do it is not presented or discussed, and no further reference to the previous paper is made.
In my opinion, the authors must revise all the manuscripts highlighting the main achievements in comparison to the previous paper and consider resubmitting the revised version.
Author Response
Answer. We agree with the reviewer. Briefly, the main differences between Pinto et al.'s study and ours are the number of GH3 genes identified and quantified in the biological model, the type of explants used, and the quantification of auxin. Pinto et al. analyzed the GH3 genes present in the genome of C. canephora and evaluated the expression profiles of four GH3 genes in non-embryogenic cells, embryogenic cell suspension, and embryogenic cells of C. arabica. In our study, we identified and evaluated the expression of the 18 GH3 genes directly throughout our SE induction process from the explants; at the same time, the free and conjugated auxin was quantified. Still, no quantification of free and conjugated auxin was performed by Pinto et al. With the above, we concluded that GH3 could be regulating the induction of SE in C. canephora through the conjugation of auxin with acidic amino acids as a substrate. For the nomenclature of GH3 in C. canephora, the 19 Arabidopsis GH3 sequences (GH3.1 to GH3.19) were downloaded from the Arabidopsis Information Resources Database (TAIR). Subsequently, blast analysis of the Arabidopsis sequences was performed against the GH3 sequences of C. canephora. Those sequences with high similarity were considered as possible GH3 homologs of A. thaliana. In general, the distribution of C. canephora gene families is very consistent compared to other gene families in plants [1]. Within the C. canephora genome, all protein-coding sequences have been grouped with 36 other plant species, classifying 24,359 (95%) sequences in 4543 clusters [1]. More than 50% of these groups were functionally annotated according to the curated catalog of gene families available on GreenPhylDB, which groups genes that code for model plant proteins such as A. thaliana and O. sativa [2]. Therefore, analyzing genes between species to identify homologies is a reliable strategy for functional annotation.
No solo el número de genes analizados, pero la interpretación del análisis y el papel que pueden jugar los productos de dichos genes en la homeostasis de las auxinas durante el proceso embriogénico.
References
[1] A. Dereeper, S. Bocs, M. Rouard, V. Guignon, S. Ravel, C. Tranchant-Dubreuil et al., The coffee genome hub: a resource for coffee genomes, Nucleic Acids Res. 43 (2015) D1028-D1035.
[2] M. Rouard, V. Guignon, C. Aluome, M.A. Laporte, G. Droc, C. Walde et al., GreenPhylDB v2.0: comparative and functional genomics in plants, Nucleic Acids Res. 39 (2011) D1095-D1102.

Reviewer 2 Report
The manuscript “Genome-wide analysis, modeling and identification of amino acid binding motifs suggest the involvement of GH3 genes during somatic embryogenesis of Coffea canephora“ (manuscript id plants-1323207) by Hugo A. Méndez-Hernández, Ana O. Quintana-Escobar, Miguel A. Uc-Chuc and Víctor M. Loyola-Vargas contains characteristic of the Gretchen Hagen 3 (GH3) genes and proteins coded by these genes from of Coffea canephora. The authors study the role of these genes in somatic embryogenesis, the process which has been well developed by the authors before.
The manuscript present the results of CcGH3 genes and protein phylogenetic analysis, gene expression analysis and protein structural studies. Authors provide also results related to CcGH3 proteins function in auxins homeostasis and somatic embryogenesis. In my opinion, the manuscript is interesting for the Plants journal also because of the importance of auxin homeostasis and the use of an interesting model in research - organogenesis. The manuscript, although written in fairly correct and understandable language (sometimes too colloquial) - can be improved by minor corrections. Some sentences needs corrections to make the text sections logically connect. Hence the suggestions that the authors would subject their manuscript to such revision, either by a person with a good command of English or by a company dealing with the revision of scientific manuscripts. I have highlighted my comments on language corrections in the document “Plants-1323207-peer-review-JP.pdf.”. Apart from the comments in the pdf version of the review, I would like to draw the authors' attention to two issues: 1 - the definition of the induction of SE, and, 2 – phylogenetic analysis of GH3 proteins.
Ad 1. The induction of somatic embryogenesis is not well defined as it is not stated whether this is a 56-day process when the circular explants are incubated in the modified Yasuda medium, or if it is a moment when plants are transferred from the pretreatment buffer to this medium.
Ad 2. Phylogenetic analysis of GH3 proteins is an important part of the manuscript - it is discussed in two chapters and presented on four figures (Fig. 3, Fig. 4, Fig. S1 and Fig. S4). However, the presented results are inconsistent, and the phylogenetic trees from individual figures have different topologies. For example – protein Cc-GH3.5 is most closely related to At-GH3.10 and At-GH3.11 (Fig. S1) which form a sister group to all other A. thaliana GH3 proteins. So you'd expect Cc-GH3.5 protein to also have a similar position to the other GH3 from C. canephora. But in Figure 3, Cc-GH3.5 is in a sub-cluster that includes Cc-GH3.9, Cc-GH3 .1a - Cc-GH3.1d. Also the positions of the Cc-GH3.3 and Cc-GH3.17i proteins on Fig. 3 and Fig. 4 are different.
In my opinion, the most valuable and reliable are the results presented on Fig. S1, where the tree is based on the largest number of sequences because using too few sequences in phylogenetic analysis causes their wrong grouping, weak clustering, etc. Because of this, I treat the results presented on Fig. S1 as a true tree. On the Fig. S1 there is clearly shown that there are C. canephora GH3 proteins closely related, or belonging, to clade III. They are the proteins from Cc-GH3.17a to Cc-GH3.17h (nine proteins). These proteins form a sister group to the proteins: At-GH3.7, At-GH3.8, and At-GH3.12 to AtGH3.19, i.e. to the clade III. The proteins from the two branches of the sister group are much more closely related to each other than to other homologs, even from the same species. This means that these proteins forms phylogenetic unite – clade. Introducing names for these clade and sub-clades would facilitate their discussion, and the most simple is to keep name “clade III” for the clade which includes both sister groups, which could be named III-A, III-B, etc.
Establishing phylogenetic relationships within the gene family is important because during evolution it is possible to preserve: 1 - the molecular function of genes or proteins (here, the ability of the GH3 protein to interact with a specific amino acid), and/or, 2 - promoter specificity resulting in conservancy of gene expression profile. Therefore, I propose that the authors re-analyze the results of their phylogenetic analyzes and ensure their consistency. When presenting trees with different topologies, authors should discuss these differences and point to their sources.
In conclusion, I recommend the manuscript plants-1323207 for Plants after improving the English language, correcting the indicated inaccuracies and the authors' responses to comments on the phylogeny of CcHG3 proteins.

Author Response
Q: Ad 1. The induction of somatic embryogenesis is not well defined as it is not stated whether this is a 56-day process when the circular explants are incubated in the modified Yasuda medium, or if it is a moment when plants are transferred from the pretreatment buffer to this medium.
Answer: We agree with the reviewer. We have clarified in the manuscript that SE induction of C. canephora is a 56-day process that begins when circular explants are incubated in Yasuda medium. In addition, we have added a supplementary figure (now Figure S5) to show the process of induction of SE graphically.
Q: Ad 2. Phylogenetic analysis of GH3 proteins is an important part of the manuscript - it is discussed in two chapters and presented on four figures (Fig. 3, Fig. 4, Fig. S1 and Fig. S4). However, the presented results are inconsistent, and the phylogenetic trees from individual figures have different topologies. For example – protein Cc-GH3.5 is most closely related to At-GH3.10 and At-GH3.11 (Fig. S1) which form a sister group to all other A. thaliana GH3 proteins. So you'd expect Cc-GH3.5 protein to also have a similar position to the other GH3 from C. canephora. But in Figure 3, Cc-GH3.5 is in a sub-cluster that includes Cc-GH3.9, Cc-GH3 .1a - Cc-GH3.1d. Also the positions of the Cc-GH3.3 and Cc-GH3.17i proteins on Fig. 3 and Fig. 4 are different.
Answer. We agree with the reviewer. In A. thaliana, 19 members of the GH3 family are divided into three groups (I, II, and III) according to their sequence similarities and substrate specificity [1]. Arabidopsis group I only includes the AtGH3.10 and GH3.11 proteins [1]. AtGH3.11 is directly related to the conjugation of isoleucine with jasmonic acid and acts as JA-amido synthetase (Figure S4) [2]. AtGH3.10 is associated with the dfl2 mutant and is involved in the elongation of the specific hypocotyl of red light, but no enzymatic activity has been reported for this protein [3]. In our work, each of the C. canephora GH3 sequences was analyzed through blasts with the A. thaliana GH3 sequences. Those GH3 sequences with high similarity were considered homologous to A. thaliana (Table 1). In our work, CcGH3.5 from C. canephora presented a high homology to GH3.11 from A. thaliana. In the combined phylogenetic tree of C. canephora and A. thaliana, we observed that CcGH3.5 clusters only with AtGH3.10 and AtGH3.11, suggesting that it possibly fulfills a similar function as JA-amino synthetase.
On the other hand, to obtain a greater insight into the structural similarity and divergence of the CcGH3 genes, we constructed a phylogenetic tree but only using sequences from C. canephora to relate it to the exon-intron distribution. We observe that they are grouped into three subgroups. This is possible because only the GH3 sequences of C. canephora were analyzed; these sequences, being a family of genes, present highly conserved regions, show a grouping of similar sequences, and represent a lineage during evolution.
Q: In my opinion, the most valuable and reliable are the results presented on Fig. S1, where the tree is based on the largest number of sequences because using too few sequences in phylogenetic analysis causes their wrong grouping, weak clustering, etc. Because of this, I treat the results presented on Fig. S1 as a true tree. On the Fig. S1 there is clearly shown that there are C. canephora GH3 proteins closely related, or belonging, to clade III. They are the proteins from Cc-GH3.17a to Cc-GH3.17h (nine proteins). These proteins form a sister group to the proteins: At-GH3.7, At-GH3.8, and At-GH3.12 to AtGH3.19, i.e. to the clade III. The proteins from the two branches of the sister group are much more closely related to each other than to other homologs, even from the same species. This means that these proteins forms phylogenetic unite – clade. Introducing names for these clade and sub-clades would facilitate their discussion, and the most simple is to keep name “clade III” for the clade which includes both sister groups, which could be named III-A, III-B, etc.
Answer. We appreciate your comment. We have worked on the manuscript and added the recommended changes for Figure S1. Our results indicate that the GH3 family presents variability in the loss or gain of homologous genes concerning other species during the evolutionary process. We agree with his comment; GH3 proteins are observed closely related to each other than other homologs, forming a phylogenetic clade. In previous works, GH3 proteins form up to five different groups in the phylogenetic tree, which represent subfamilies already described, while two of them are reported as new [4].
References
[1] P.E. Staswick, B. Serban, M. Rowe, I. Tiryaki, M.T. Maldonado, M.C. Maldonado et al., Characterization of an Arabidopsis enzyme family that conjugates amino acids to indole-3-acetic acid, Plant Cell 17 (2005) 616-627.
[2] P.E. Staswick, I. Tiryaki, The oxylipin signal jasmonic acid is activated by an enzyme that conjugates it to isoleucine in Arabidopsis, Plant Cell 16 (2004) 2117-2127.
[3] T. Takase, M. Nakazawa, A. Ishikawa, K. Manabe, M. Matsui, DFL2, a new member of the Arabidopsis GH3 gene family, is involved in red light-specific hypocotyl elongation, Plant Cell Physiol. 44 (2003) 1071-1080.
[4] J.M. Vielba, Identification and initial characterization of a new subgroup in the GH3 gene family in woody plants, J. Plant Biochem. Biotechnol. 28 (2019) 280-290.

Reviewer 3 Report
In this manuscript, Mendez-Hernandez and colleagues report quantification of IAA ana IAA conjugates during C. canephora embryogenesis and GH3 analysis during this phase.
Overall idea of this manuscript deserves merit; however authors must address several modifications.
The most important of all is that authors must deeply compare their findings with Pinto et al paper (reference 34). Pinto et al paper mostly analyse gene expression in C. arabica. How their findings relate to present findings in C. canephora? In terms of contribution to a plant science audience, I see that is much more important to compare diploid/tetraploid coffee findings rather than comparing results with model plants or other species that are distant from coffee.
In lines 100-102 authors say that the major drawback in that paper is that they did not relate gene notation with A. thaliana genes. Nowadays, this kind of relation brings more problems than solutions. For instance, in lines 188-198, authors claim that for some A. thaliana genes, C. canephora homologs have "more than one copy". That's exactly why people nowadays do not follow strictly A. thaliana notations when they report a genome-wide analysis in other plant; it is well known that each genome will have an unique duplication pattern that will not fit well in A. thaliana notation.
In this sense, it will be much wiser to authors to compare differences of annotation between their analysis and Pinto et al. analyses. Pinto et al report 17 GH3 genes in C. canephora and this study report 18 genes. Why authors find this difference? This would be an important topic to be explored.
In Results (lines 111-125) and Methods (lines 444-459), authors do not report which C. canephora genotype (cultivar) was used for embryogenesis. Since it is well known that there are genotypic differences of competence, this is a mandatory information.
Lines 138-142 fit better in Introduction rather than in results.
In figure 2, authors must use a statistical test to detect differences. With the exception of IAA-Glu, all other forms of IAA are not obvious to be analyzed by eye.
Author should also make a correlation analysis to address: is there any GH3 gene that have a transcriptional profile minimally correlated to IAA content?
line 182: it is not clear why and how authors used UGENE to find GH3 genes.
Table 1: I do not agree with notations attached to A. thaliana genes, but this is much more a matter of personal view of science rather than a methodological problem. However, in this table authors should use accession numbers of A. thaliana genes instead of simple notation.
In Phylogenetic analyses, Table 2 should be moved to supplementary material, since it does not bring any important information to the paper. In this table, authors should consider change "median total length" to "genome size" (although I do not think this is an important information). There are also some typos in the table (i.e. some species with 0, others with --; Setaria italic)
Figure 3 is less informative than Figure S1. Figure 3 should be supressed, and figure S1 should be brought to the main paper. Authors must also insert group information directly in the phylogenetic tree.
Items 2.7 and 2.8 should be merged in a single topic. Authors do not explain in methods about how they did figure 5. I understood it come from a previous study; however, it must be explained here. Since data from figure 5 is from another study, authors should consider bring figures S2 and S3 to the main paper and move figure 5 to the supplementary material.
Authors should also discuss if RNA-seq data matches qPCR data, and a statistical test in qPCR data is also welcome. More importantly, qPCR data is possible of correlation with IAA quantification? Authors should include this comparison.
Methods:
Lines 476-477: Was the efficiency of primers assessed? how? Was cyclophilin gene used as a reference in a previous study? Was it recommended for this kind of experiment? Authors must report it.
Lines 480-481: its is not clear how authors initially obtained GH3 genes. Did authors made a keyword search? Blast? Domain analysis? This must be reported. The Coffee Genome Hub has an associated paper (https://doi.org/10.1093/nar/gku1108) that must be cited.
In phylogenetic analysis (lines 487-490), it is mandatory that authors use a maximum likelihood approach, incluing selection of protein evolution models and gap treatment. This would be an important improvement of Pinto et al. analyses.
There are also numbering problems in supplementary material that authors must address.
There are several minor corrections that could be pointed in this manuscript; however it is important to address these major changes firstly.
Author Response
Q: The most important of all is that authors must deeply compare their findings with Pinto et al paper (reference 34). Pinto et al paper mostly analyse gene expression in C. arabica. How their findings relate to present findings in C. canephora? In terms of contribution to a plant science audience, I see that is much more important to compare diploid/tetraploid coffee findings rather than comparing results with model plants or other species that are distant from coffee.
Answer. Thank you for your suggestion. We have added some information comparing the main differences between Pinto’s paper and ours in the Discussion section. Regarding the ploidy, what is known so far is that different numbers of ploidy influence the embryogenic response of the different species belonging to the genus Coffea. The average number of cotyledonary somatic embryos was different according to the level of ploidy of the species, even under homogeneous culture conditions during the induction of indirect SE [1]. Thus, the induction of SE in the different species is influenced by factors such as epigenetic changes, conditions of incubation, the composition of the medium, and the type of explant used. In order to make a fair comparison between the ploidy level and SE, it is recommended to investigate DNA methylation thoroughly, as it is known that methylation has a role in gene regulation and consequently in plant growth and development processes [1,2].
Q: In lines 100-102 authors say that the major drawback in that paper is that they did not relate gene notation with A. thaliana genes. Nowadays, this kind of relation brings more problems than solutions. For instance, in lines 188-198, authors claim that for some A. thaliana genes, C. canephora homologs have "more than one copy". That's exactly why people nowadays do not follow strictly A. thaliana notations when they report a genome-wide analysis in other plant; it is well known that each genome will have an unique duplication pattern that will not fit well in A. thaliana notation.
Answer. We appreciate your comment. Our rationale is that the distribution of GH3 genes varies significantly between different plant species. Polyploidization may play a crucial role in the evolution of plant genomes [3]. Moreover, gene duplication is considered a crucial mechanism in the evolutionary history of plants. Plant diversification is greatly favored by gene duplication events, resulting in the generation of new genes necessary for the evolution of plants [4]. The distribution of C. canephora gene families is very consistent compared to other gene families in plants [5]. Within the C. canephora genome, all protein-coding sequences have been grouped with 36 other plant species, classifying 24,359 (95%) sequences in 4,543 clusters [5]. More than 50% of these groups were functionally annotated according to the curated catalog of gene families available on GreenPhylDB, which groups genes that code for model plant proteins such as A. thaliana and O. sativa [6].
Q: In this sense, it will be much wiser to authors to compare differences of annotation between their analysis and Pinto et al. analyses. Pinto et al report 17 GH3 genes in C. canephora and this study report 18 genes. Why authors find this difference? This would be an important topic to be explored.
Answer. We appreciate the suggestion. We have added this discussion to the Materials and Methods section. Apparently, in the work of Pinto et al. (2019), the GH3 genes were numbered consecutively as their sequences were analyzed, without subsequently making any homology comparison with A. thaliana. In our work, for the nomenclature of GH3 in C. canephora, the 19 Arabidopsis GH3 sequences (GH3.1 to GH3.19) were downloaded from the Arabidopsis Information Resources Database (TAIR). Subsequently, blast analysis of the Arabidopsis sequences was performed against the GH3 sequences of C. canephora. We identified all the sequences of the GH3 genes; a search was carried out within the reference genome of C. canephora available online (http://coffee-genome.org/) [5]. A total of 20 candidate sequences were found. Since the GH3 family consists of characteristic domains, all recovered GH3 sequences were analyzed in the NCBI conserved domain database (https://www.ncbi.nlm.nih.gov/Structure/cdd/wrpsb-cgi). Eighteen sequences coincided with the GH3 domain (pfam03321 and PLN2247) and were considered candidate GH3 sequences.
Q: In Results (lines 111-125) and Methods (lines 444-459), authors do not report which C. canephora genotype (cultivar) was used for embryogenesis. Since it is well known that there are genotypic differences of competence, this is a mandatory information.
Answer. We appreciate your comment and have we have added this information.
Q: Lines 138-142 fit better in Introduction rather than in results.
Answer. We appreciate your comment. We have taken it into account.
ANSWER. In figure 2, authors must use a statistical test to detect differences. With the exception of IAA-Glu, all other forms of IAA are not obvious to be analyzed by eye.
Answer. We appreciate your comment. Statistical analysis was added to the figure.
Q: Author should also make a correlation analysis to address: is there any GH3 gene that have a transcriptional profile minimally correlated to IAA content?
Answer. Thank you for your observation. We have now added some information to the Discussion section that compares the IAA content, RNAs-seq analysis, and qPCR results.
Q: line 182: it is not clear why and how authors used UGENE to find GH3 genes.
Answer. Thanks for the observation. The sequences of the GH3 genes were downloaded into the reference genome of C. canephora available online (http://coffee-genome.org/) in the "downloads" section. In order to view the downloaded file, the use of the UGENE Software was required, which is available for download (http://ugene.net/).
Q: Table 1: I do not agree with notations attached to A. thaliana genes, but this is much more a matter of personal view of science rather than a methodological problem. However, in this table authors should use accession numbers of A. thaliana genes instead of simple notation.
Answer. We appreciate your comment. We have made the recommended changes.
Q: In Phylogenetic analyses, Table 2 should be moved to supplementary material, since it does not bring any important information to the paper. In this table, authors should consider change "median total length" to "genome size" (although I do not think this is an important information). There are also some typos in the table (i.e. some species with 0, others with --; Setaria italic)
Answer. We appreciate your comment. We have moved Table 2 to the supplemental material. We have also made the recommended changes.
Q: Figure 3 is less informative than Figure S1. Figure 3 should be supressed, and figure S1 should be brought to the main paper. Authors must also insert group information directly in the phylogenetic tree.
Answer. We appreciate your observation. We have inserted the group information into the phylogenetic tree. We performed a phylogenetic analysis between C. canephora and A. thaliana; our results show that the GH3 proteins of C. canephora are only present in groups I and II. There was no relationship with any of group III. Our comparison with A. thaliana is consistent with previous reports in other species such as Malus sieversii [7], Zea mays [8], Cicer arietinum, Medicago truncatula and Lotus japonicus [9]. All these phylogenetic analyses showed clusters only in groups I and II.
Q: Items 2.7 and 2.8 should be merged in a single topic. Authors do not explain in methods about how they did figure 5. I understood it come from a previous study; however, it must be explained here. Since data from figure 5 is from another study, authors should consider bring figures S2 and S3 to the main paper and move figure 5 to the supplementary material.
Answer. We appreciate your observation and have added the corresponding information to Materials and Methods section 4.5.
On the other hand, we would like to add that although the RNA-seq data from the SE induction process were reported in another study by our group, it is in this work that the specific analysis of GH3 transcripts was carried out, so we would like to keep, if it is possible, the figure within the main manuscript.
Q: Authors should also discuss if RNA-seq data matches qPCR data, and a statistical test in qPCR data is also welcome. More importantly, qPCR data is possible of correlation with IAA quantification? Authors should include this comparison.
Answer. One of the most notable results when comparing the quantification data of free and conjugated auxin and the analysis of relative expression is that one day after the start of the induction stage, there is a considerable decrease in the amount of free auxin. At the same time, the amount of IAA-Ala and IAA-Leu conjugates increases, and the IAA-Gl conjugate decreases. At the same points, the relative expression of specific GH3 genes also increases. These results suggest regulation of the auxin concentration through conjugation gives rise to SE.
Methods:
Q: Lines 476-477: Was the efficiency of primers assessed? how? Was cyclophilin gene used as a reference in a previous study? Was it recommended for this kind of experiment? Authors must report it.
Answer. Goulao et al. [10] carried out a comparative study between different internal reference genes, identifying the most stable ones for quantitative real-time PCR gene expression in Coffea spp. We used the cyclophilin gene as an internal reference based on previous studies. Subsequently, within our research group, three candidates were evaluated, of which cyclophilin was selected for the validation of transcriptomic data by qPCR in C. canephora [11]. We have added the appropriate citation in section 4.4.
Q: Lines 480-481: its is not clear how authors initially obtained GH3 genes. Did authors made a keyword search? Blast? Domain analysis? This must be reported. The Coffee Genome Hub has an associated paper (https://doi.org/10.1093/nar/gku1108) that must be cited.
Answer. We agree with the reviewer and have made the changes that he recommends. We have added more information on how C. canephora GH3 genes were obtained in this study in section 4.5. We have also added the recommended reference.
Q: In phylogenetic analysis (lines 487-490), it is mandatory that authors use a maximum likelihood approach, incluing selection of protein evolution models and gap treatment. This would be an important improvement of Pinto et al. analyses.
Answer. We appreciate your observation. We have added the information in section 4.6.
Q: There are also numbering problems in supplementary material that authors must address.
Answer. We appreciate your valuable comment and have corrected this mistake.
References
[1] N.A. Sanglard, P.M. Amaral-Silva, M.C. Sattler, S.C. de Oliveira, L.M. Cesário, A. Ferreira et al., Indirect somatic embryogenesis in Coffea with different ploidy levels: a revisiting and updating study, Plant Cell Tissue Organ Cult. 136 (2019) 255-267.
[2] H. Zhang, A. Ali, F. Hou, T. Wu, D. Guo, X. Zeng et al., Effects of ploidy variation on promoter DNA methylation and gene expression in rice (Oryza sativa L.), BMC Plant Biol. 18 (2018) 1-12.
[3] J.F. Wendel, S.A. Jackson, B.C. Meyers, R.A. Wing, Evolution of plant genome architecture, Genome Biol. 17 (2016) 37.
[4] M.R. Mckain, H. Tang, J.R. McNeal, S. Ayyampalayam, J.I. Davis, C.W. dePamphilis et al., A phylogenomic assessment of ancient polyploidy and genome evolution across the Poales, Genome Biology and Evolution 8 (2016) 1150-1164.
[5] A. Dereeper, S. Bocs, M. Rouard, V. Guignon, S. Ravel, C. Tranchant-Dubreuil et al., The coffee genome hub: a resource for coffee genomes, Nucleic Acids Res. 43 (2015) D1028-D1035.
[6] M. Rouard, V. Guignon, C. Aluome, M.A. Laporte, G. Droc, C. Walde et al., GreenPhylDB v2.0: comparative and functional genomics in plants, Nucleic Acids Res. 39 (2011) D1095-D1102.
[7] H. Yuan, K. Zhao, H. Lei, X. Shen, Y. Liu, X. Liao et al., Genome-wide analysis of the GH3 family in apple (Malus x domestica), BMC Genomics 14 (2013) 297.
[8] S. Feng, R. Yue, S. Tao, Y. Yang, L. Zhang, M. Xu et al., Genome-wide identification, expression analysis of auxin-responsive GH3 family genes in maize (Zea mays L.) under abiotic stresses, J. Int. Plant Biol. 57 (2015) 783-795.
[9] V.K. Singh, M. Jain, R. Garg, Genome-wide analysis and expression profiling suggest diverse roles of GH3 genes during development and abiotic stress responses in legumes, Front. Plant Sci. 5 (2015) 789.
[10] L.F. Goulao, A.S. Fortunato, J.C. Ramalho, Selection of reference genes for normalizing quantitative real-time PCR gene expression data with multiple variables in Coffea spp, Plant Mol. Biol. Rep. 30 (2012) 741-759.
[11] A.O. Quintana-Escobar, G.I. Nic-Can, R.M. Galaz-Ávalos, V.M. Loyola-Vargas, E. Góngora-Castillo, Transcriptome analysis of the induction of somatic embryogenesis in Coffea canephora and the participation of arf and AUX/IAA genes, PeerJ 7 (2019) e7752.

Reviewer 4 Report
This manuscript reports on GH3 family genes in C. canephora. Somatic embryogenesis in species like coffee is important to understand and improve, therefore this is a relevant study. They found 18 genes within the genome, did some phylogenetic analysis to compare to Arabidopsis, characterized expression by RNA-seq (confirmed by qRT-PCR for some members), examined IAA and IAA-conjugate accumulation during somatic embryogenesis in coffee, and examined predicted structure of the GH3 proteins.
There is a lot of work in this manuscript. What hurts is that some of it was previously published by Pinto et al. in 2019. We all get “scooped” sooner or later and it is painful (been there). Pinto et al. did some genome wide analysis of GH3 in the same species, but their phylogenetic trees seems to categorize Cc genes in different categories (for example, they have gene in class III). I realize antagonism is not the best way to go, but I wonder if this group has the same genes and, if so, why or why not they think their category scheme is better or not (as in Pinto et al. I think have some in category III and this paper does not)? Pinto et al. also did some structure analysis, and characterization of expression in embryogenic and non-embryogenic cells. So I think a key to this manuscript is to emphasize what is new and different. I also firmly believe confirmation in science is critical as well but maybe not for Plants. They do characterize transcript during the SE process and measure free IAA and conjugated IAA amounts. I think this is novel.
Figure 2 – please indicate significance on the graph. Some, based on error bars should be, but I know I have been surprised. Also, I do not understand what “ 3 bio reps of at least 2 indep exps” means (figure legend). Are not bio reps totally independent experiments? Do they mean 3 technical reps from 2 bio exps?
I agree with the scheme to use similar nomenclature with other species. But often, first published is what sticks. Maybe a table to compare Pinto et al. genes with theirs and Arabidopsis (and any other species)? This could be supplemental.
Table 1. – unless you have shown localization, I would add “predicted” to Subcell. Loc. (and provide the reference as different algorithms give different results). I would also suggest adding I, II (or III) here.
Section 2.7 – how many reps of RNA-seq and significance? Has the RNA-seq data been deposited (or plans to be) in a public database?
Section 2.8, line 276 – they say Table S3… do they mean Table S1 and/or Fig S2?
Author Response
Q: Pinto et al. did some genome wide analysis of GH3 in the same species, but their phylogenetic trees seems to categorize Cc genes in different categories (for example, they have gene in class III). I realize antagonism is not the best way to go, but I wonder if this group has the same genes and, if so, why or why not they think their category scheme is better or not (as in Pinto et al. I think have some in category III and this paper does not)?
Answer. We agree with the reviewer. In A. thaliana, 19 members of the GH3 family are divided into three groups (I, II, and III) according to their sequence similarities and substrate specificity [1]. The Arabidopsis group I GH3 proteins are JA-amido synthetases [2]. The Arabidopsis group, II GH3 proteins, are involved in the conjugation of IAA with various amino acids [1]. At present, group III proteins have only been identified in Arabidopsis, AtGH3-12 / PBS3, which catalyzes conjugation between benzoates and 4-substituted amino acids and participates in SA signaling [3]. We performed a phylogenetic analysis between C. canephora and A. thaliana and determined that the GH3 proteins of C. canephora are only present in groups I and II. There was no relationship with any of group III. This comparison with A. thaliana is consistent with previous reports in other species such as Malus sieversii [4], Zea mays [5], Cicer arietinum, Medicago truncatula and Lotus japonicus [6] where the phylogenetic analysis showed clusters only in groups I and II.
In the phylogenetic tree of Pinto et al., 2019, there is confusion regarding groups I, II, and III previously reported in Arabidopsis [1]. They mention that group III of the phylogenetic tree has the most studied members and all grouped A. thaliana proteins associated with the conjugation of amino acids with auxin. This in fact is the opposite. Only members of group II have the potential to conjugate auxin with amino acids. It is also likely that some sequences do not correspond to members of group II. It is possible that of the four CcGH3 that Pinto et al. (2019) selected as possible candidates involved in conjugation, only one has the conserved residues for the amino acid binding sites and auxin.
Pinto et al. also did some structure analysis, and characterization of expression in embryogenic and non-embryogenic cells. So I think a key to this manuscript is to emphasize what is new and different. I also firmly believe confirmation in science is critical as well but maybe not for Plants. They do characterize transcript during the SE process and measure free IAA and conjugated IAA amounts. I think this is novel.
Q: Figure 2 – please indicate significance on the graph. Some, based on error bars should be, but I know I have been surprised. Also, I do not understand what “ 3 bio reps of at least 2 indep exps” means (figure legend). Are not bio reps totally independent experiments? Do they mean 3 technical reps from 2 bio exps?
Answer. We agree with the reviewer. Statistical analysis was added to the figure. We appreciate your comment and proceeded to change the abbreviation as it is certainly confusing. We meant that three independent biological replicates were used.
Q: I agree with the scheme to use similar nomenclature with other species. But often, first published is what sticks. Maybe a table to compare Pinto et al. genes with theirs and Arabidopsis (and any other species)? This could be supplemental.
Answer. We appreciate the suggestion. Apparently, in the work of Pinto et al. (2019), the GH3 genes were numbered consecutively as their sequences were analyzed, without subsequently making any homology comparison with A. thaliana. Those sequences with high similarity were considered as possible GH3 homologs of A. thaliana. We use the nomenclature regarding the GH3 of A. thaliana, also based on the literature. In general, the distribution of C. canephora gene families is very consistent compared to other gene families in plants [7]. Within the C. canephora genome, all protein-coding sequences have been grouped with 36 other plant species, classifying 24,359 (95%) sequences in 4,543 clusters [7]. More than 50% of these groups were functionally annotated according to the curated catalog of gene families available on GreenPhylDB, which groups genes that code for model plant proteins such as A. thaliana and O. sativa [8]. Therefore, analyzing genes between species to identify homologies is a reliable strategy for functional annotation.
Q: Table 1. – unless you have shown localization, I would add “predicted” to Subcell. Loc. (and provide the reference as different algorithms give different results). I would also suggest adding I, II (or III) here.
Answer. We agree with the reviewer, and the change has been added to Table 1. The subcellular location prediction reference is shown in Materials and Methods.
Q: Section 2.7 – how many reps of RNA-seq and significance? Has the RNA-seq data been deposited (or plans to be) in a public database?
Answer. We appreciate your comments. The RNA-seq analysis was previously performed and reported by our group [9]. RNA-seq data of our SE induction process in C. canephora is available at NCBI with GEO accession number GSE128888. We have added this information to the manuscript in section 2.7.
Q: Section 2.8, line 276 – they say Table S3… do they mean Table S1 and/or Fig S2?
Answer. We agree with the reviewer and have corrected the error.
References
[1] P.E. Staswick, B. Serban, M. Rowe, I. Tiryaki, M.T. Maldonado, M.C. Maldonado et al., Characterization of an Arabidopsis enzyme family that conjugates amino acids to indole-3-acetic acid, Plant Cell 17 (2005) 616-627.
[2] P.E. Staswick, I. Tiryaki, M.L. Rowe, Jasmonate response locus JAR1 and several related Arabidopsis genes encode enzymes of the firefly luciferase superfamily that show activity on jasmonic, salicylic, and indole-3-acetic acids in an assay for adenylation, Plant Cell 14 (2002) 1405-1415.
[3] R.A. Okrent, M.D. Brooks, M.C. Wildermuth, Arabidopsis GH3.12 (PBS3) conjugates amino acids to 4-substituted benzoates and is inhibited by salicylate, J. Biol. Chem. 284 (2009) 9742-9754.
[4] H. Yuan, K. Zhao, H. Lei, X. Shen, Y. Liu, X. Liao et al., Genome-wide analysis of the GH3 family in apple (Malus x domestica), BMC Genomics 14 (2013) 297.
[5] S. Feng, R. Yue, S. Tao, Y. Yang, L. Zhang, M. Xu et al., Genome-wide identification, expression analysis of auxin-responsive GH3 family genes in maize (Zea mays L.) under abiotic stresses, J. Int. Plant Biol. 57 (2015) 783-795.
[6] V.K. Singh, M. Jain, R. Garg, Genome-wide analysis and expression profiling suggest diverse roles of GH3 genes during development and abiotic stress responses in legumes, Front. Plant Sci. 5 (2015) 789.
[7] A. Dereeper, S. Bocs, M. Rouard, V. Guignon, S. Ravel, C. Tranchant-Dubreuil et al., The coffee genome hub: a resource for coffee genomes, Nucleic Acids Res. 43 (2015) D1028-D1035.
[8] M. Rouard, V. Guignon, C. Aluome, M.A. Laporte, G. Droc, C. Walde et al., GreenPhylDB v2.0: comparative and functional genomics in plants, Nucleic Acids Res. 39 (2011) D1095-D1102.
[9] A.O. Quintana-Escobar, G.I. Nic-Can, R.M. Galaz-Ávalos, V.M. Loyola-Vargas, E. Góngora-Castillo, Transcriptome analysis of the induction of somatic embryogenesis in Coffea canephora and the participation of arf and AUX/IAA genes, PeerJ 7 (2019) e7752.

Round 2
Reviewer 3 Report
In this version, authors addressed many issues.
However, the paper still need another round of revision since I still raise some issues:
- authors can maintain figure 5 in the paper; however, it is mandatory to bring figures S2 and S3 to the main text. Since qPCR was done in this article, it should be detailed accordingly in the main text.
- line 680: The sentence :"The analysis was conducted using the Neighbor-Joining method." must be removed. Authors do not have clarity about phylogenetic analyses. A NJ is only the starting point; however, the overall method is maxiumum likelihood.
- Authors did not answered about how they accessed primer efficiency. This is a mandatory issue, since authors did not corrected their relative expression by efficiency. It is well-known that this issue can influence in results interpretation (https://www.sciencedirect.com/science/article/pii/S2214753514000059, https://doi.org/10.1016/j.ymeth.2012.08.011, https://academic.oup.com/clinchem/article/67/6/829/6247760). Authors might use Miner (http://miner.ewindup.info/, https://link.springer.com/article/10.1007/s10658-019-01747-6 ) or LinRegPCR (https://medischebiologie.nl/files/, https://www.ncbi.nlm.nih.gov/pmc/articles/PMC7238024/, https://www.nature.com/articles/srep28348 ).
- section 4.4 must cite references of Goulao et al. and Quintana-Escobar et al.
- the order in methods should reflect the order of presented data. In this sense, sections 4.9 and 4.10 should be presented before RNA extraction. Moreover, authors must report statistical analyses in methods.
After these changes, the paper can be re-analyzed.
Author Response
Review 3 Q: "In this version, authors addressed many issues. However, the paper still need another round of revision since I still raise some issues: authors can maintain figure 5 in the paper; however, it is mandatory to bring figures S2 and S3 to the main text. Since qPCR was done in this article, it should be detailed accordingly in the main text. Answer. We agree with the reviewer. We have made the recommended changes. Q: line 680: The sentence :"The analysis was conducted using the Neighbor-Joining method." must be removed. Authors do not have clarity about phylogenetic analyses. A NJ is only the starting point; however, the overall method is maxiumum likelihood. Answer. We appreciate your comment. We have made the suggested change. Q: Authors did not answered about how they accessed primer efficiency. This is a mandatory issue, since authors did not corrected their relative expression by efficiency. It is well-known that this issue can influence in results interpretation (https://www.sciencedirect.com/science/article/pii/S2214753514000059, https://doi.org/10.1016/j.ymeth.2012.08.011, https://academic.oup.com/clinchem/article/67/6/829/6247760). Authors might use Miner (http://miner.ewindup.info/, https://link.springer.com/article/10.1007/s10658-019-01747-6 ) or LinRegPCR (https://medischebiologie.nl/files/, https://www.ncbi.nlm.nih.gov/pmc/articles/PMC7238024/, https://www.nature.com/articles/srep28348) Answer. First of all, we would like to offer an apology for not adequately answer this question in the previous review, as we did not fully understand the context surrounding it. Next, we thank the reviewer for asking such an important question that highlights a point often overlooked when performing real-time quantitative PCR. We agree with your concern as we are aware of the importance of guaranteeing the efficiency of PCR; thanks to your advice, we learned to use one of the recommended software to verify and confirm that the efficiency was within the allowed ranges. We chose the LinRegPCR software [1] as it included within its interface the Applied Biosystem Step One equipment that we used for real-time PCR. As a result, we obtained the efficiency value for the primers. For example, the efficiency value was 1.903, 1.911, 1.91, 1.91, 1.905, and 1.903 for GH3.17a, GH3.17c, GH3.17h, GH3.3b, GH3.1c, and Cyclophilin primers, respectively. This software takes a value of 2 as 100% of efficiency. Without a doubt, from now on, it will be a fundamental part of our future studies. Also, we would like to mention a series of factors that we took into account before and during the performance of the quantitative PCR to guarantee the highest possible efficiency: Due to the availability of the C. canephora genome [2], specific primers were designed for each gene of interest, considering that it is a multiethnic family, the candidate sequences were aligned with MEGA 7 (www.megasoftware.net). The primers were designed to avoid conserved regions between the genes of interest to decrease nonspecificity. The primers were designed at NCBI (https://www.ncbi.nlm.nih.gov/tools/primer-blast/), taking into account different parameters in the design. The length of the primers was established between 20 and 30 bp; a GC content between 45 and 60% was considered, and a melting temperature between 59 °C and 60 °C. The amplification ranges of the amplicon were established between 150 and 210 bp. In addition, the probability of secondary structure formation and dimerization was verified in the Oligo Explorer 1.4 software (http://www.genelink.com/tools/gl-oe.asp). In silico PCR tests were performed on the primers designed to verify amplification using the Sol Genomics Network online tool (https://solgenomics.net/tools/in_silico_pcr). We performed end-point PCR with the respective controls and a temperature gradient to confirm the optimal melting temperature of the designed primers. Additionally, the equipment incorporates an "Efficiency" column in the results table, given on a scale of 0-1, where 1 = 100% efficiency. The equipment specifications and some manual recommendations were taken into account to guarantee efficiencies, such as the fine selection of the internal reference gene and the quantification method. The reference gene selection was based on previous studies where its effectiveness and stability within a group of genes were evaluated. The reference gene selection was based on previous studies where its effectiveness and stability within a group of genes were evaluated. We indirectly evaluated the efficiency. This evaluation was made qualitatively through the visualization of the amplification curve provided by the equipment as a result, which takes into account the cycles and fluorescence values, verifying that the geometric slopes were parallel. The above is under the assumption that slopes that are not parallel indicate less than 100% efficiency. We reiterate our appreciation for your observation as we have undoubtedly learned how to use accurate, novel, user-friendly software, which will be helpful from now on to give robustness to our analyses. Q: section 4.4 must cite references of Goulao et al. and Quintana-Escobar et al. Answer. We agree with the reviewer. We have added the corresponding references. Q: the order in methods should reflect the order of presented data. In this sense, sections 4.9 and 4.10 should be presented before RNA extraction. Moreover, authors must report statistical analyses in methods." Answer. We agree with the reviewer. We have made the recommended changes. References 1. Ramakers, C., Ruijter, J.M., Deprez, R.H.L., Moorman, A.F. Assumption-free analysis of quantitative real-time polymerase chain reaction (PCR) data. Neuroscience letters 2003, 339, 62-66, https://doi.org/10.1016/S0304-3940(02)01423-4. 2. Denoeud, F., Carretero-Paulet, L., Dereeper, A., Droc, G., Guyot, R., Pietrella, M., Zheng, C., Alberti, A., Anthony, F., Aprea, G. et al. The coffee genome provides insight into the convergent evolution of caffeine biosynthesis. Science 2014, 345, 1181-1184, http://doi.org/10.1126/science.1255274.

Reviewer 4 Report
The authors have sufficiently addressed my comments. Thank you.
Author Response
A professional English editor has reviewed the manuscript.
Round 3
Reviewer 3 Report
Authors addressed the most important issues. I only strongly recommend a readability and grammar analysis of the manuscript. However, most experimental problems were solved.
Author Response
The English of this paper, like that of all my papers, was proofread by Emily Wortman-Wunder, Assistant Professor, English Department, University of Colorado Denver.